JCB Journal of Cell Biology

# TLNRD1 is a CCM complex component and regulates endothelial barrier integrity

Neil J. Ball[1,2]*, Sujan Ghimire[3]*, Gautier Follain[3,4], Ada O. Pajari[3], Diana Wurzinger[3], Monika Vaitkevičiūtė[3], Alana R. Cowell[1], Bence Berki[4], Johanna Ivaska[4,5,6,7,8], Ilkka Paatero[4], Benjamin T. Goult[1,2], and Guillaume Jacquemet[3,4,8,9]

**We previously identified talin rod domain-containing protein 1 (TLNRD1) as a potent actin-bundling protein in vitro. Here, we report that TLNRD1 is expressed in the vasculature in vivo. Its depletion leads to vascular abnormalities in vivo and modulation of endothelial cell monolayer integrity in vitro. We demonstrate that TLNRD1 is a component of the cerebral cavernous malformations (CCM) complex through its direct interaction with CCM2, which is mediated by a hydrophobic C-terminal helix in CCM2 that attaches to a hydrophobic groove on the four-helix domain of TLNRD1. Disruption of this binding interface leads to CCM2 and TLNRD1 accumulation in the nucleus and actin fibers. Our findings indicate that CCM2 controls TLNRD1 localization to the cytoplasm and inhibits its actin-bundling activity and that the CCM2-TLNRD1 interaction impacts endothelial actin stress fiber and focal adhesion formation. Based on these results, we propose a new pathway by which the CCM complex modulates the actin cytoskeleton and vascular integrity.**

## Introduction

The actin cytoskeleton is essential for most cellular processes, including cell movement, cell division, and cell shape maintenance. Consequently, aberrant actin cytoskeleton regulation is linked to multiple illnesses, including cancer and immunological and neurological disorders. Actin dynamics are regulated by various proteins, which include nucleators, polymerizers, depolymerizers, and crosslinkers, all working together to ensure spatially and temporally appropriate assembly of actin structures (Lappalainen et al., 2022). While the role of individual actin regulatory proteins is starting to be understood in cells, their functions in living organisms often remain elusive.

We previously identified talin rod domain-containing protein 1 (TLNRD1, also known as MESDC1) as a potent actin-bundling protein in vitro and cultured cells (Cowell et al., 2021). We found that TLNRD1 is homologous to the R7R8 domains of the cytoskeletal adaptor talin (Gingras et al., 2010) and exists as a constitutive homodimer (Cowell et al., 2021). By solving the TLNRD1 structure, we demonstrated that TLNRD1 comprises a four-helix domain (TLNRD1[4H], equivalent to talin R8) inserted into a five-helix domain (TLNRD1[5H], equivalent to talin R7) to form a nine-

helix module and that TLNRD1 homodimerization is mediated via an interface located on the four-helix module. Like talin R7R8, TLNRD1 binds F-actin, but because TLNRD1 forms an antiparallel dimer, it also bundles F-actin. In cancer cells, TLNRD1 localizes to the cytoplasm and accumulates on actin bundles and in filopodia (Cowell et al., 2021). Functionally, we reported that TLNRD1 expression enhanced filopodia formation and cancer cell migration, while TLNRD1 downregulation had the opposite effect. Other studies have shown that TLNRD1 overexpression is associated with increased proliferation and xenograft growth in hepatocellular carcinoma (Wu et al., 2017) and that TLNRD1 depletion reduced bladder cancer cell migration and invasion (Tatarano et al., 2012). While TLNRD1 appears to modulate cancer cell functions, little is known about the physiological functions of TLNRD1 in normal tissue.

The cerebral cavernous malformations (CCM) complex is a trimeric protein assembly critical for vascular homeostasis. This complex comprises three proteins: Krev interaction trapped protein 1 (KRIT1 or CCM1), cerebral cavernous malformations 2 protein (CCM2), and programmed cell death protein 10

[1]School of Biosciences, University of Kent, Canterbury, UK; [2]Department of Biochemistry, Cell and Systems Biology, Institute of Systems, Molecular and Integrative Biology, University of Liverpool, Liverpool, UK; [3]Faculty of Science and Engineering, Cell Biology, Åbo Akademi University, Turku, Finland; [4]Turku Bioscience Centre, University of Turku and Åbo Akademi University, Turku, Finland; [5]Department of Life Technologies, University of Turku, Turku, Finland; [6]Western Finnish Cancer Center (FICAN West), University of Turku, Turku, Finland; [7]Foundation for the Finnish Cancer Institute, Helsinki, Finland; [8]InFLAMES Research Flagship Center, University of Turku and Åbo Akademi University, Turku, Finland; [9]Turku Bioimaging, University of Turku and Åbo Akademi University, Turku, Finland.

*N.J. Ball and S. Ghimire contributed equally to this paper. Correspondence to Guillaume Jacquemet: guillaume.jacquemet@abo.fi; Benjamin T. Goult: b.t.goult@liverpool.ac.uk; Ilkka Paatero: ilanpa@utu.fi

A.O. Pajari's current affiliation is Institute of Biotechnology, University of Helsinki, Helsinki, Finland. A.R. Cowell's current affliation is Institute of Structural and Molecular Biology, Birkbeck College, London, UK.

(PDCD10 or CCM3). Together, they orchestrate endothelial cell functions by modulating endothelial cell junctions and the actin cytoskeleton. The CCM complex is pivotal in regulating the MEKK3–MEK5–ERK5 signaling cascade and the small GTPase RhoA (Su and Calderwood, 2020). Specifically, by regulating RhoA activity, the CCM complex facilitates actin remodeling, which is indispensable for effective cell migration and underpins endothelial cell junction robustness. Importantly, disruptions in the functions of these proteins, whether through mutations or loss of function, are intrinsically linked to CCM disease, a neurovascular disorder characterized by the emergence of blood-filled cavernomas within the central nervous system (Riolo et al., 2021).

Here, we report that TLNRD1 is a member of the CCM complex. In zebrafish embryos, silencing of TLNRD1 results in vascular malformations, while in endothelial cells, it disrupts monolayer integrity and actin organization. We found that TLNRD1 directly binds to CCM2 through a hydrophobic interaction involving a C-terminal helix in CCM2 and a groove on TLNRD1's 4-helix domain. As our findings demonstrate that CCM2 inhibits TLNRD1's actin-binding activity, we propose that the CCM complex can also modulate the actin cytoskeleton and vascular integrity by controlling TLNRD1 activity.

## Results

### TLNRD1 is a putative member of the CCM complex
We previously described TLNRD1 as an actin-bundling protein contributing to filopodia formation in cancer cells (Cowell et al., 2021). However, the broad localization of TLNRD1 in cells suggests it likely has other roles. To further understand TLNRD1 cellular functions, we sought to identify TLNRD1 binding partners using an unbiased mass-spectrometry approach. We performed GFP pulldowns from cells expressing either GFP or GFP-TLNRD1 followed by mass spectrometry analysis (Fig. 1, A and B). Using this strategy, we identified 89 proteins as specifically enriched to TLNRD1 pulldowns (Fig. 1, A and B; and Table S1). Interestingly, mapping putative TLNRD1 binders onto a protein–protein interaction network revealed that several hits, namely KRIT1 (also known as CCM1), CCM2, PDCD10 (also known as CCM3), serine/threonine kinase 25 (STK25), and integrin β1-binding protein 1 (ITGB1BP1, also known as ICAP1) cluster together (Fig. 1 C). These proteins are known to assemble the CCM complex, a trimeric protein complex of KRIT1–CCM2–PDCD10, with STK25 and ITGB1BP1 being accessory members (Yang et al., 2023; Faurobert et al., 2013). In addition, western blot analyses confirmed that KRIT1 and ITGB1BP1 copurify with GFP–TLNRD1, further validating our mass spectrometry analyses (Fig. 1 D). These results led us to speculate that TLNRD1 could be a full, or accessory, member of the CCM complex.

We previously showed that TLNRD1 and ITGB1BP1 accumulate at the tips of myosin-X (MYO10)-induced filopodia in cancer cells (Jacquemet et al., 2019; Cowell et al., 2021). Therefore, we next assessed, using structured illumination microscopy, the ability of the other CCM complex members to localize to filopodia tips (Fig. S1). These experiments revealed that CCM2 and

PDCD10 could also be found at the tip of MYO10-induced filopodia (Fig. S1), further linking TLNRD1 to the CCM complex.

### TLNRD1 depletion leads to vascular malformation in vivo
While the individual CCM complex components are expressed in diverse cell types, the CCM complex has a well-documented role in forming and maintaining the vasculature. Indeed, loss of function mutations of KRIT1, CCM2, or PDCD10 leads to CCM, which are vascular lesions defined in patients by blood-filled endothelial cell caverns and an absence of a mature vessel wall (Chohan et al., 2019).

Analysis of publicly available single-cell RNA-Seq datasets (Tabula Muris [Schaum et al., 2018]) revealed that in adult mice, TLNRD1 is expressed in endothelial cells in the brain (Fig. 2 A). TLNRD1 expression is, however, not limited to the brain vasculature, and TLNRD1 transcripts were also detected in other organs, for instance, in the endothelial cells in the heart as well as other cell types including fibroblasts and leukocytes (Fig. 2 B). To validate these datasets, we stained and imaged mouse brain slices and observed that TLNRD1 was expressed in PECAM-positive vessels (Fig. 2 C). Together these data indicate that TLNRD1 is expressed in the vascular endothelium in vivo.

Zebrafish embryos are robust model organisms for studying the cardiovascular system (Hogan and Schulte-Merker, 2017). In addition, mutation of CCM complex components, including *krit1*, *ccm2*, or *pdcd10*, in zebrafish embryos leads to defects in the vasculature (Yoruk et al., 2012; Hogan et al., 2008). Therefore, we next investigated the impact of *tlnrd1* gene disruption in zebrafish embryos using CRISPR (Fig. S2 A). Strikingly, zebrafish embryos treated with anti-TLNRD1 CRISPR guide RNAs exhibited severe abnormal morphologies at multiple anatomical sites. In particular, the mesencephalic, mid-cerebral, and caudal vein plexus were dilated in TLNRD1-targeted embryos (Fig. 2, D and E). These phenotypes were not observed in the control embryos. TLNRD1-targeted embryos also had lower heart rates than control embryos, but these results did not reach statistical significance (Fig. S2 B). Interestingly, single-cell RNA-Seq datasets revealed that TLNRD1 expression peaks at 24 h after fertilization during zebrafish embryo development, where it is principally expressed in the blood vasculature (Zebrahub [Lange et al., 2023, *Preprint*]). Similarly, TLNRD1 is also expressed in the vasculature in human embryos (Fig. S2 C [Xu et al., 2023]). Importantly, a reanalysis of publicly available datasets revealed that TLNRD1 expression at the RNA level is upregulated in CCM lesions in patients (Fig. S2 D, Subhash et al., 2019). While the observed effects of TLNRD1 depletion in zebrafish embryos could be attributed to TLNRD1 functions beyond the vascular system, our findings suggest a significant role for TLNRD1 in vascular regulation in vivo.

### TLNRD1 modulates endothelial monolayer integrity
Zebrafish embryos are powerful tools for studying and observing the development of the cardiovascular system; however, they are less amenable to mechanistic studies. Therefore, we next investigated the contribution of TLNRD1 to endothelial cell functions in cellulo. Human umbilical vein endothelial cells (HUVEC) express TLNRD1 (Fig. S3 A) (Cowell et al., 2021). In

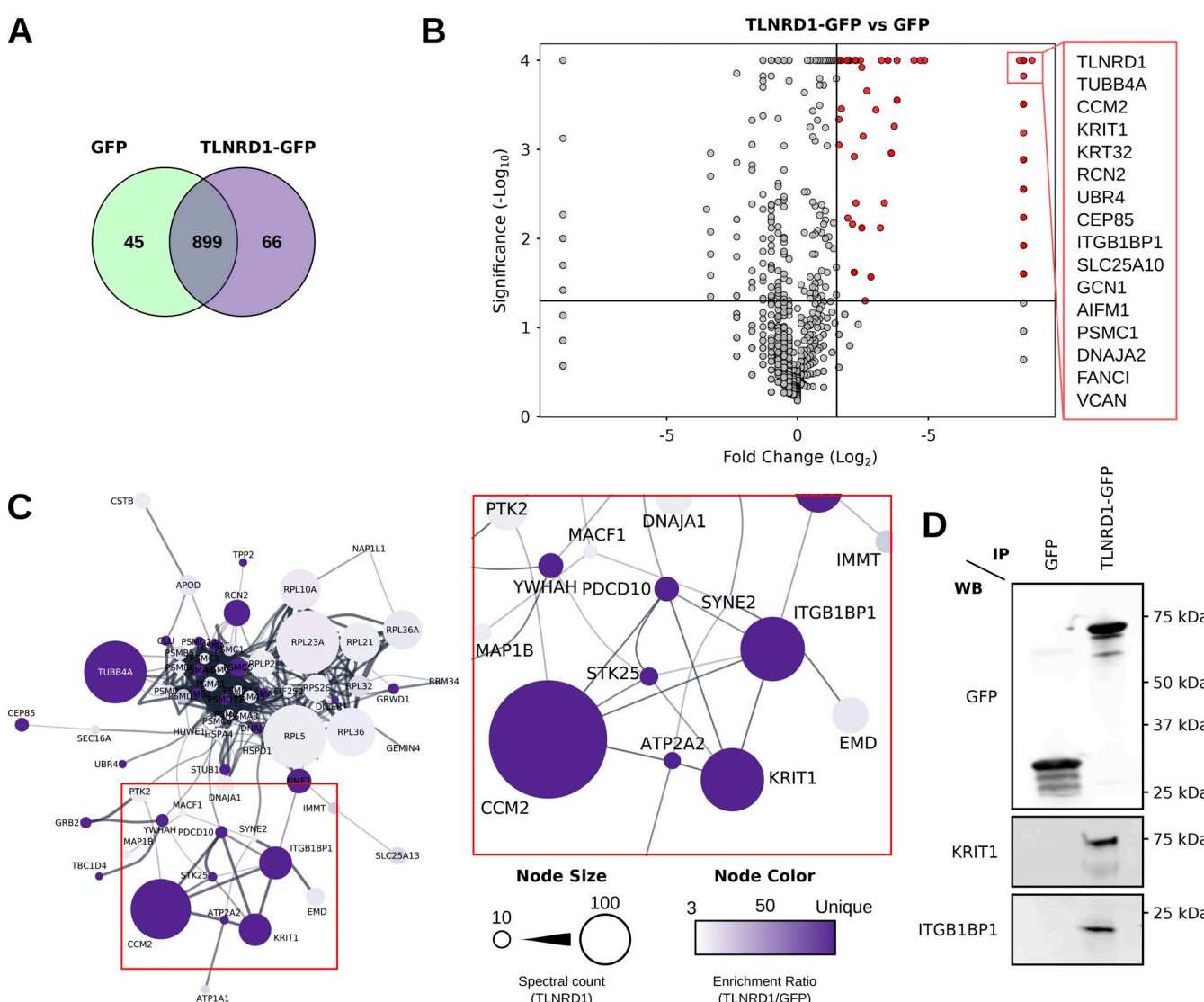

Figure 1. **Mass spectrometry analyses identify TLNRD1 as a putative member of the CCM complex. (A–C)** Mass spectrometry (MS) analysis of GFP-TLNRD1-binding proteins. A comparison of the GFP-TLNRD1 and GFP datasets is displayed in Venn diagram (A) and volcano plot (B). In the volcano plot, the enrichment ratio (TLNRD1 over GFP) for each protein detected is plotted against the significance of the association (see Table S1 for the MS data). Notably, proteins uniquely identified in either the TLNRD1 or GFP conditions were assigned a fold change of 400 to be displayed on the volcano plot. **(C)** Proteins specifically enriched to TLNRD1 were mapped onto a protein–protein interaction network (STRING, see the Materials and methods for details). Each node (circle) represents a protein (labeled with gene name), and each edge (line) represents a reported interaction between two proteins. The node's color indicates the enrichment ratio of that particular protein (TLNRD1 over GFP). The node's area represents the spectral count of that specific protein in the TLNRD1-GFP dataset. **(D)** GFP-pulldown in HEK293T cells expressing GFP-TLNRD1 or GFP alone. KRIT1 and ITGB1BP1 recruitment to the bait proteins was then assessed by Western blotting (representative of three biological repeats). Source data are available for this figure: SourceData F1.

HUVEC monolayers, TLNRD1 localized to the cytoplasm. It accumulated on actin bundles, including stress fibers and filopodia (Fig. 3 A). TLNRD1 also localized on the actin structures in the vicinity of cell–cell junctions (Fig. 3 A). However, we did not observe an accumulation of TLNRD1 at cell–cell junctions.

Next, we assessed the contribution of TLNRD1 to endothelial monolayer integrity. TLNRD1 expression was silenced using two independent siRNAs (Fig. 3 B). After 3 days, the resulting monolayers were fixed, stained for fibronectin without permeabilization, and imaged. This approach allowed us to assess the integrity of the endothelial monolayer by quantifying the size and number of fibronectin patches, as these patches on the

ventral side of the monolayer become accessible to the antibody when the junction above them is leaky (Fig. S3 B). Notably, we did not detect any fibronectin patches on the apical surface. Our results demonstrate that silencing TLNRD1 disrupts monolayer integrity at the observed time point, irrespective of whether flow stimulation was applied (Fig. 3, C and D; and Fig. S3 C). When imaged at higher magnification, we observed that TLNRD1-silenced cells were more spread out than control cells (Fig. 3, E and F). In addition, the overall organization of the actin cytoskeleton in the formed monolayers appeared altered. In TLNRD1-silenced cells, actin stress fibers were more prominent on individual cells' edges, rendering the overall

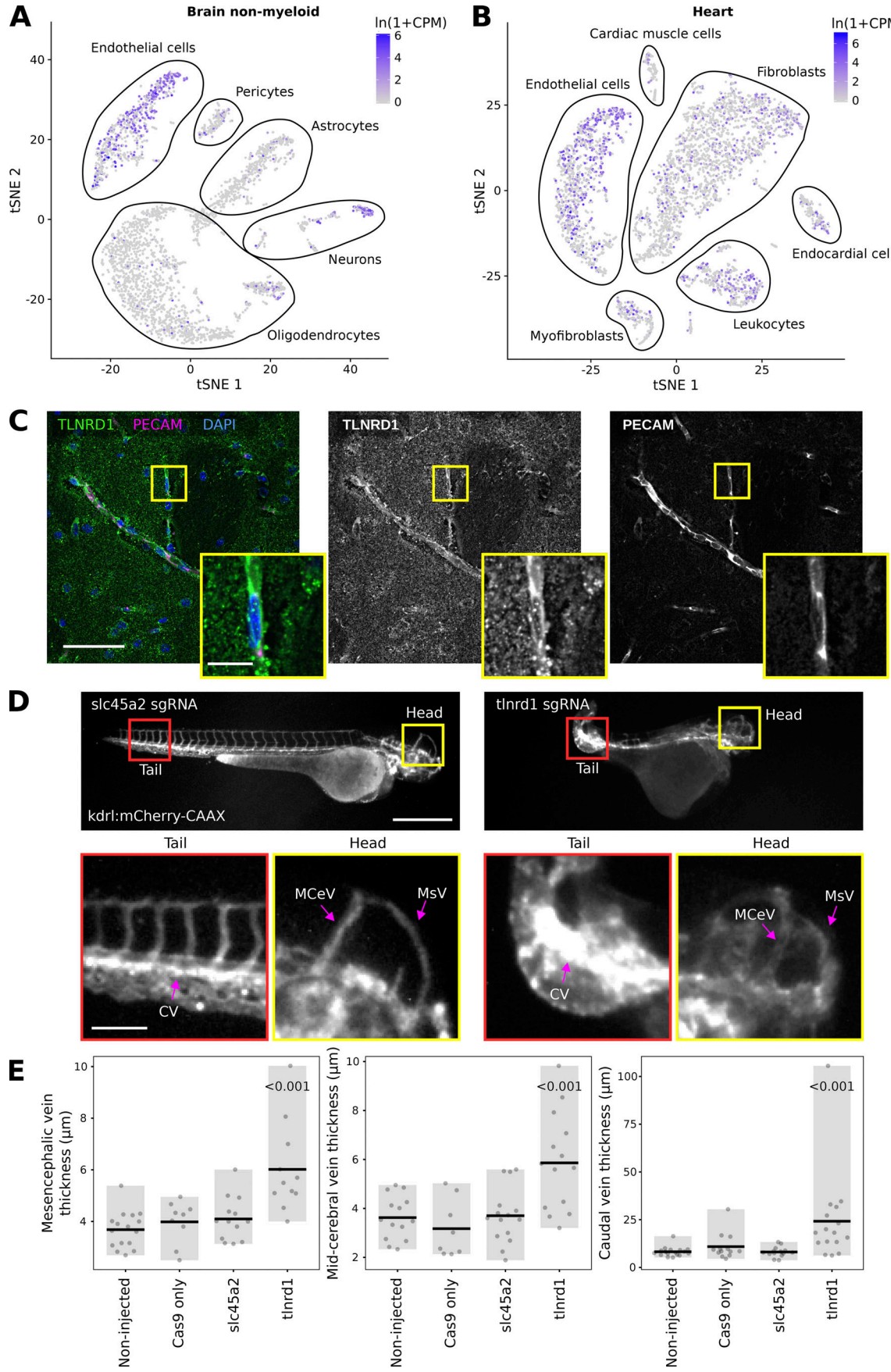

Figure 2. **TLNRD1 is expressed in endothelial cells in vivo and regulates the vascular system. (A and B)** TLNRD1 expression in mouse brain (A) and mouse heart (B). This single-cell RNA-Seq data is from the Tabula Muris dataset (Schaum et al., 2018). For the brain, endothelial cells were defined as Cdh5+, Pecam1+, Slco1c1+, and Ocln+; in the heart, endothelial cells were defined as Cdh5+ and Pecam1+ (Schaum et al., 2018). **(C)** Mouse brain slices were stained for TLNRD1, PECAM, and DAPI and imaged using a spinning disk confocal microscope. A single Z-plane is displayed. The yellow square highlights a magnified region of interest (ROI). Scale bars: (main) 50 µm and (inset) 10 µm. **(D and E)** kdrl:mCherry-CAAX zebrafish embryos were injected with recombinant Cas9 alone or together with sgRNA targeting TLNRD1 or slc45a2. The embryos were then imaged using a fluorescence microscope. **(D)** Representative images are displayed. The yellow and red squares highlight ROIs, which are magnified. The mesencephalic (MsV), mid-cerebral (MCeV), and caudal (CV) vein plexus are highlighted. Scale bars: (main) 500 µm and (inset) 100 µm. **(E)** The thickness of the mesencephalic, mid-cerebral, and caudal vein plexus measured from microscopy images are plotted as dot plots (non-injected, $n = 17$; Cas9, $n = 13$; slc45a2, $n = 14$; TLNRD1, $n = 16$). The gray bar highlights the data distribution, while the black line indicates the mean. The P values were determined using a randomization test.

actin organization in the monolayer more organized (Fig. 3, E and G). Yet, TLNRD1 silencing did not affect phospho-myosin light chain levels (Fig. 3 H; and Fig. S4, A and B).

Next, we measured the transendothelial electrical resistance (TEER) during monolayer formation. This assay measures the electrical resistance across a cellular monolayer and thus can report on the monolayer integrity and permeability. As the monolayer forms, a maximal TEER is reached. Repeating these experiments following TLNRD1 silencing revealed a delay in reaching maximal TEER, indicating defects in assembling an impermeable monolayer (Fig. 3, I and J). Finally, we assessed the contribution of TLNRD1 in modulating endothelial barrier function following the establishment of an impermeable monolayer. To investigate this, we treated the established monolayers with thrombin, a protease known to induce endothelial barrier dysfunction and mimic inflammatory conditions. Interestingly, consistent with previous findings (Schnitzler et al., 2024), TLNRD1 silencing consistently enhanced endothelial barrier function after thrombin treatment (Fig. S4, C and D), underscoring its critical role in modulating endothelial permeability. Altogether, our data indicate that TLNRD1 contributes to overall actin cytoskeleton organization in endothelial cells and modulates both the establishment and regulation of endothelial monolayer permeability; it contributes to forming an impermeable monolayer initially, but subsequently, TLNRD1 modulates the permeability in the established monolayer.

### TLNRD1 binds to CCM2 via its four-helix bundle

Having shown that TLNRD1 copurifies with the CCM complex (Fig. 1) and that TLNRD1 depletion leads to vascular phenotypes in both zebrafish embryos (Fig. 2) and endothelial cells (Fig. 3), we next wanted to map the interaction(s) between TLNRD1 and the CCM complex. We implemented a protein trapping strategy to identify which CCM protein(s) interacts with TLNRD1. CCM2 and PDCD10 were targeted to the mitochondria, and the ability of TLNRD1 to be recruited to this compartment was then analyzed using fluorescence microscopy (Fig. 4, A and B). Using this strategy, we found that TLNRD1 strongly colocalized with mitochondrial-targeted (mito)-CCM2 but not mito-PDCD10 (Fig. 4, A and B). Interestingly, TLNRD1 and mito-CCM2 clustered strongly together, forming aggregates (Fig. 4 A). Significantly, the recruitment of TLNRD1 to mito-CCM2 was not affected when KRIT1 expression was silenced using siRNA (Fig. S5, B and C), indicating that the TLNRD1–CCM2 interaction does not require KRIT1. Next, we expressed TLNRD1 and CCM2 in endothelial cells. We found that both proteins colocalize on

actin structures and in the cytosol (Fig. 4 C). Our data indicate that TLNRD1 interacts with the CCM complex via CCM2.

To map the TLNRD1 domain interacting with CCM2, we performed GFP-trap experiments in cells expressing GFP, GFP-TLNRD1, GFP-TLNRD1$^{4H}$, or GFP-TLNRD1$^{5H}$ (Fig. 4 D). CCM2 coprecipitated with TLNRD1 and TLNRD1$^{4H}$ but not TLNRD1$^{5H}$, indicating that CCM2 interacts with TLNRD1 via the TLNRD1 4-helix bundle (Fig. 4 D). Next, we used a GST-pulldown assay with recombinant proteins to determine whether the TLNRD1–CCM2 interaction is direct (Fig. S5 D). GST-CCM2 copurified with recombinant TLNRD1 and TLNRD1$^{4H}$ but not with TLN1$^{R7R8}$ (Fig. S5 D), despite TLN1$^{R7R8}$ and TLNRD1$^{4H}$ sharing a similar domain organization and structure (Cowell et al., 2021). Our data indicate that TLNRD1 directly interacts with CCM2 via the TLNRD1 four-helix bundle domain.

### The TLNRD1–CCM2 interaction involves the C-terminal helix in CCM2

CCM2 contains two characterized domains, an N-terminal phosphotyrosine binding (PTB) domain and a C-terminal harmonin homology domain (HHD) (Fig. 4 E). To map the region of CCM2 responsible for interacting with TLNRD1, we generated several CCM2 truncated constructs designed based on the known CCM2 structures (Fisher et al., 2013, 2015; Wang et al., 2015) and the AlphaFold CCM2 structure prediction (Jumper et al., 2021). Using GST-pulldown and recombinant proteins, we found that TLNRD1 copurifies in vitro with CCM2, CCM2$^{283-444}$, and CCM2$^{413-438}$ but not with CCM2$^{1-231}$ or CCM2$^{283-379}$ (Fig. 4 F). These assays were only qualitative, so to quantify this interaction, we used a Fluorescence Polarization (FP) assay, in which unlabeled TLNRD1 was titrated against a fixed concentration of fluorescein-labeled CCM2(413-438) peptide. The FP assay revealed that TLNRD1 and TLNRD1$^{4H}$ interact with CCM2$^{413-438}$ with a dissociation constant ($K_d$) in the nanomolar range (Fig. 4 G). At the same time, no interaction between CCM2$^{413-438}$ and TLN1$^{R7R8}$ was detected (Fig. 4 G). Our data indicate that CCM2 does not bind TLNRD1 via the previously characterized CCM2 HHD or PTB domains. Instead, CCM2 interacts with TLNRD1 via the CCM2$^{413-438}$ region, which AlphaFold predicts as a helix, which we call here the CCM2 C-terminal helix (CTH) for simplicity.

### The TLNRD1–CCM2 binding interface involves a hydrophobic groove on TLNRD1 and hydrophobic residues of CCM2

Having mapped the regions in CCM2 and TLNRD1 responsible for their interaction, we next modeled the CCM2-TLNRD1 complex using ColabFold (Mirdita et al., 2022) (Fig. 5 A). The

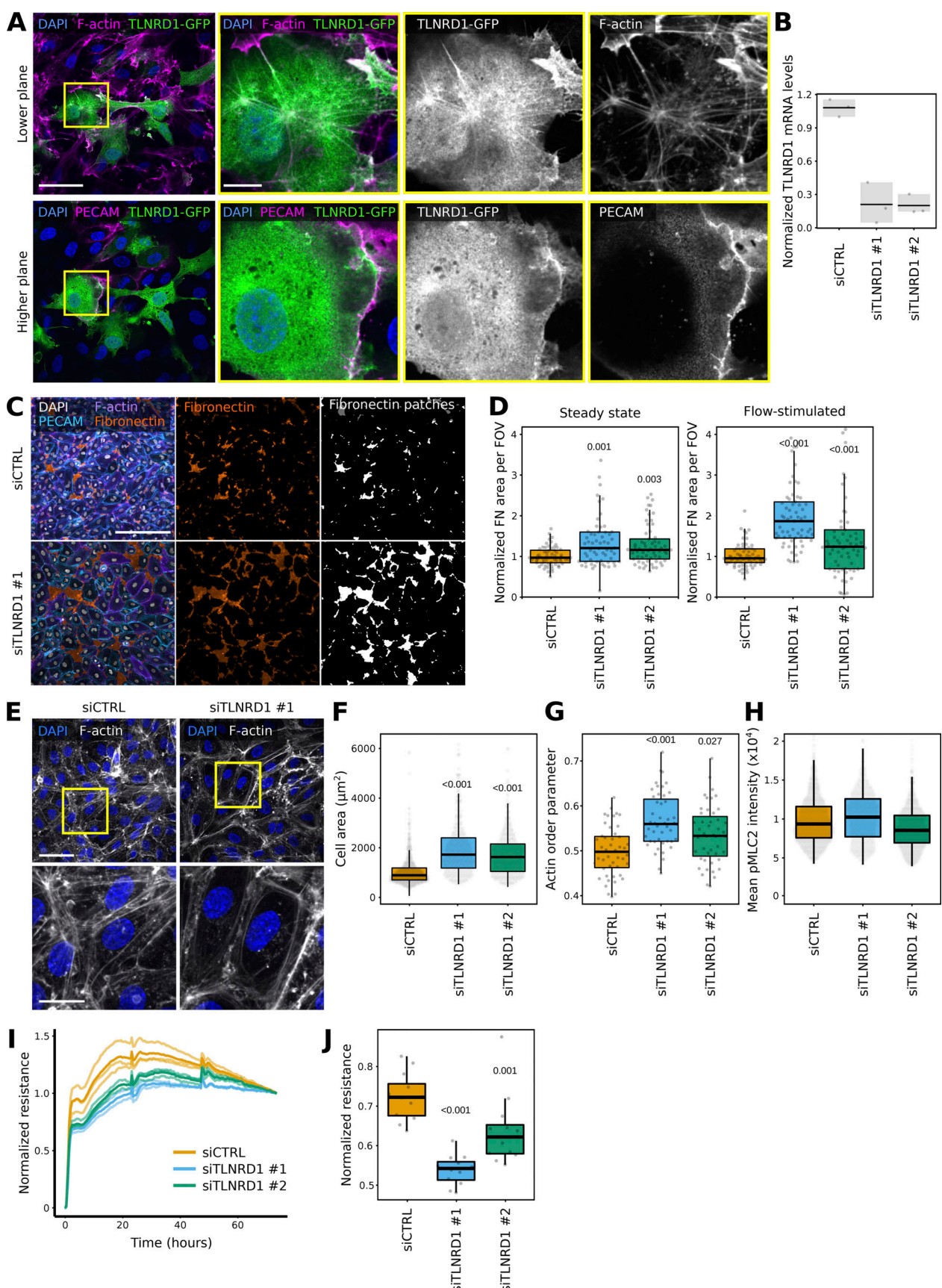

Figure 3. **TLNRD1 modulates endothelial monolayer integrity. (A)** HUVECs expressing TLNRD1-GFP were fixed, stained for DAPI, F-actin, and PECAM, and imaged using a spinning disk confocal microscope. Two Z-planes from the same field of view are displayed. The yellow squares highlight magnified ROIs. Scale

bars: (main) 50 µm and (inset) 10 µm. **(B–D)** TLNRD1 expression was silenced in HUVECs using two independent siRNA. **(B)** TLNRD1 expression levels were determined by qPCR. **(C and D)** HUVEC cells were allowed to form a monolayer in the presence or absence of flow stimulation. Cells were then fixed and stained for DAPI, F-actin, PECAM, and fibronectin (without permeabilization) before imaging on a spinning disk confocal microscope. **(C)** Representative maximum intensity projections are displayed (flow stimulation). Scale bar: 250 µm. **(D)** The area covered by fibronectin patches in each field of view was then quantified (three biological repeats, n > 60 fields of view per condition). **(E–H)** TLNRD1 expression was silenced in HUVECs using two independent siRNAs, and cells were allowed to form a monolayer without flow stimulation. Cells were then fixed and stained for DAPI and F-actin or phospho-Myosin light chain (pMLC S20). Images were acquired using a spinning disk confocal microscope. **(E)** Representative SUM projections are displayed. Scale bar: (main) 50 µm and (inset) 20 µm. **(F)** The cell area was measured using manual cell segmentation (three biological repeats, >45 fields of view, >460 cells per condition). **(G)** The actin organization (order parameter) was quantified using Alignment by Fourier Transform (Marcotti et al., 2021). **(H)** The average pMLC intensity per cell is displayed. In this case, cells were automatically segmented using cellpose (three biological repeats, n > 861 cells per condition). **(I and J)** Assessment of trans-endothelial electrical resistance (TEER) in siCTRL and siTLNRD1 endothelial monolayers was conducted utilizing the xCELLigence system. Individual TEER trajectories were normalized to their final readings to study the establishment of the TEER over time. **(I)** Displays represent data from one biological replicate. Here, the mean TEER trajectory from three individual wells is delineated with a bold line. In contrast, individual TEER curves are rendered in a lighter shade to delineate specific measurements within the same replicate. **(J)** Focuses on the comparative analysis at the time when siCTRL cells attain 70% of their ultimate TEER values, highlighting the impact of TLNRD1 silencing on developing endothelial barrier function (4 biological repeats, 11 measurements). The results are shown as Tukey boxplots. The whiskers (shown here as vertical lines) extend to data points no further from the box than 1.5× the interquartile range. The P values were determined using a randomization test.

ColabFold prediction suggests that CCM2$^{CTH}$ binds to TLNRD1$^{4H}$, with each TLNRD1 monomer capable of binding one CCM2$^{CTH}$ (Fig. 5 A and Video 1) in agreement with our biochemical analysis (Fig. 4 H). Analysis of the predicted binding interface indicated that the TLNRD1–CCM2 interaction was predominantly hydrophobic, with the hydrophobic residues of the amphipathic CCM2$^{CTH}$ inserted into a hydrophobic groove on the surface of TLNRD1$^{4H}$ (Fig. 5 B). To test this model, we designed targeted point mutations in TLNRD1 and in CCM2 to perturb their interaction. Two CCM2 mutants were designed, namely CCM2$^{I432D}$ and CCM2$^{W421D/D422A}$ (termed here CCM2$^{WD/AA}$). Using the FP assay, we found that the CCM2$^{WD/AA}$ and CCM2$^{I432D}$ mutations abolish the TLNRD1–CCM2 $^{CTH}$ interaction in vitro, validating our structural modeling and the critical importance of the hydrophobic interface in the interaction (Fig. 5, D and E). Interestingly, an I428S mutation in CCM2 has been reported in a patient diagnosed with vascular dementia (CCM2$^{I428S}$ [Mönkäre et al., 2021]), and this isoleucine is on the same face of the CTH helix as the mutations we designed, so we predicted it too would impact the TLNRD1–CCM2 binding interface. Testing the CCM2$^{I428S}$ mutant in the FP assay showed that it had a similar effect to the CCM2$^{WD/AA}$ and CCM2$^{I432D}$ mutations, disrupting the TLNRD1–CCM2$^{CTH}$ interaction (Fig. 5 D).

We next designed mutations in TLNRD1 aiming at disrupting CCM2 binding; these double mutations, TLNRD1$^{L191T/A225T}$ (termed here TLNRD1$^{2T}$) and TLNRD1$^{K192E/R233E}$ (termed here TLNRD1$^{2E}$), were introduced into TLNRD1-4H. Importantly, the TLNRD1$^{2T}$ mutant was designed to mimic the surface of TLN1$^{R8}$ (Fig. 5 C), which does not bind to CCM2 (Fig. 4 G). We found that the TLNRD1$^{2E}$ abolished binding to CCM2$^{CTH}$ (Fig. 5 E), and the TLNRD1$^{2T}$ mutant reduced the TLNRD1 affinity for CCM2 by 150-fold (Fig. 5 E). Next, we validated our in vitro experiments in cells using the previously described mitochondrial trapping strategy. TLNRD1 or CCM2 was targeted to the mitochondria, and the ability of various TLNRD1 and CCM2 constructs to be recruited to this compartment was then analyzed using fluorescence microscopy (Fig. 5 G). As expected, we found that CCM2 strongly colocalized with mito-TLNRD1 and that TLNRD1 strongly colocalized with mito-CCM2 (Fig. 5, F and G). However, deletion of CCM2$^{CTH}$ (CCM2$^{\Delta CTH}$), or the CCM2$^{I428S}$, CCM2$^{WD/AA}$,

and CCM2$^{I432D}$ mutations all abolished CCM2 recruitment to mito-TLNRD1 (Fig. 5, F and G). In addition, both the TLNRD1$^{2E}$ and TLNRD1$^{2T}$ double mutations abolished TLNRD1 recruitment to mito-CCM2. Altogether, our mutagenesis approach demonstrates that the TLNRD1–CCM2 interaction involves a hydrophobic groove on TLNRD1 and hydrophobic residues of CCM2.

## CCM2 modulates TLNRD1 localization, actin binding, and bundling activity

After identifying point mutations that disrupt the interaction between CCM2 and TLNRD1, we explored the functions of the TLNRD1–CCM2 complex. We started by overexpressing the CCM2 mutants in endothelial cells. Surprisingly, deletion of CCM2$^{CTH}$ (CCM2$^{\Delta CTH}$), or CCM2$^{WD/AA}$, and CCM2$^{I432D}$ mutations lead to a strong accumulation of CCM2 in the nucleus (Fig. 6, A and B). This finding is particularly noteworthy because, although CCM2 is known to localize to the nucleus, the mechanisms controlling the nuclear translocation remain elusive (Swamy and Glading, 2022). A similar effect was seen with the patient mutation, CCM2$^{I428S}$, suggesting it has a loss of function phenotype in cells. However, silencing TLNRD1 did not result in any noticeable changes in CCM2 subcellular localization (Fig. 6, C and D), suggesting that the C-terminal helix in CCM2 likely serves functions beyond binding to TLNRD1. These findings complicate using these specific CCM2 mutants as tools for dissecting the role of the TLNRD1–CCM2 interaction. In addition, TLNRD1 silencing did not affect phospho-myosin light chain levels (Fig. 3 H; and Fig. S4, A and B) and decreased rather than increased KLF4 expression in endothelial cells (Fig. 6 E), both pathways regulated by CCM2 (Cuttano et al., 2016; Stockton et al., 2010).

Next, we overexpressed our engineered TLNRD1 mutants in endothelial cells and conducted a detailed analysis of their subcellular localization (Fig. 7 A). Intriguingly, both TLNRD1$^{2T}$ and TLNRD1$^{2E}$ demonstrated a subtle yet noticeable increase in nuclear localization when compared with TLNRD1$^{WT}$ (Fig. 7 B). Most remarkably, TLNRD1$^{2E}$ failed to localize to actin stress fibers. At the same time, TLNRD1$^{2T}$ exhibited a slightly enhanced localization to these structures compared with TLNRD1$^{wt}$ (Fig. 7 C).

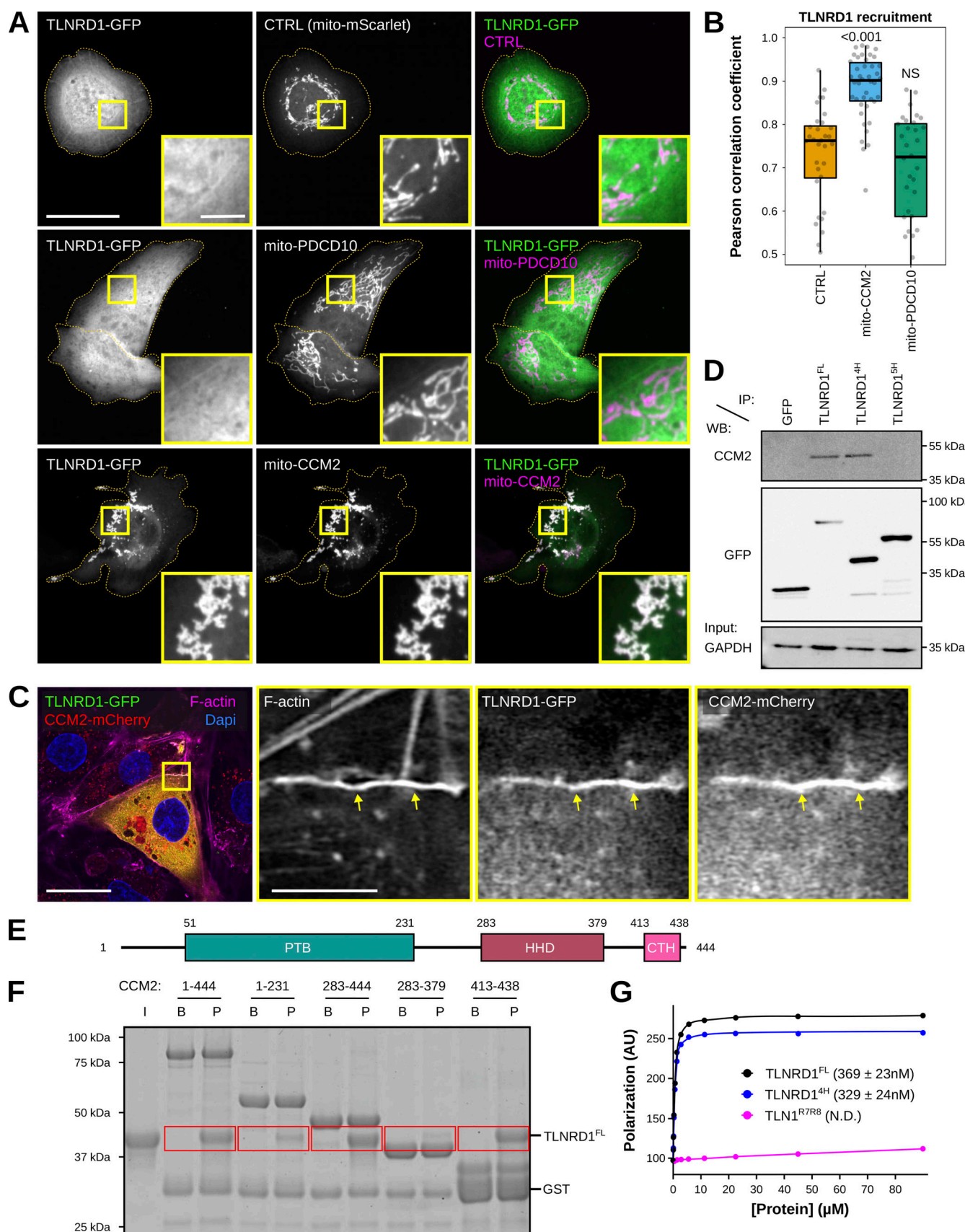

**Figure 4.** **The TLNRD1–CCM2 interaction involves the TLNRD1 4-helix bundle and a C-terminal helix in CCM2. (A and B)** U2OS cells expressing TLNRD1-GFP and mito-mScarlet (CTRL), mito-PDCD10-mScarlet, or mito-CCM2-mScarlet were imaged using a spinning disk confocal microscope. **(A)** Representative

single Z-planes are displayed. Dashed yellow lines highlight the cell outlines. The yellow squares highlight magnified ROIs. Scale bars: (main) 25 µm and (inset) 5 µm. **(B)** 3D colocalization analysis was performed using the JACoP Fiji plugin (three biological repeats, $n > 31$ image stacks per condition). The results are shown as Tukey boxplots. The whiskers (shown here as vertical lines) extend to data points no further from the box than 1.5× the interquartile range. The P values were determined using a randomization test. NS indicates no statistical difference between the mean values of the highlighted condition and the control. **(C)** HUVECs expressing TLNRD1-GFP and CCM2-mCherry were stained for F-actin and DAPI and imaged using an Airyscan confocal microscope. A single Z-plane is displayed. The yellow squares highlight a magnified ROI. Scale bars: (main) 25 µm and (inset) 5 µm. **(D)** GFP-pulldown in HEK293T cells expressing GFP-TLNRD1, GFP-TLNRD1[4H]. GFP-TLNRD1[5H] or GFP alone. CCM2 recruitment to the bait proteins was assessed by western blotting (representative of three biological repeats). **(E)** CCM2 schematic showing the boundaries of the phosphotyrosine binding (PTB) domain, the harmonin homology domain (HHD), and the C-terminal helix (CTH). **(F)** A GST-pulldown assay was used where Glutathione agarose-bound GST-CCM2 fragments (beads: B) were incubated with recombinant TLNRD (input: I). After multiple washes, proteins bound to the beads (pellet: P) were eluted. Red boxes highlight areas of interest in the gel. **(G)** A fluorescence polarization assay was used to determine the $K_d$ of the interaction between TLNRD1, TLNRD1[4H], or TLN1[R7R8] with SUMO-CCM2[CTH]. $K_d$ values (nM) are shown in parentheses. ND, not determined. Source data are available for this figure: SourceData F4.

These observations led us to hypothesize that the mutations we introduced in TLNRD1 could affect TLNRD1's ability to bind to actin. Therefore, we carried out in vitro actin pulldown assays. These experiments showed that while both TLNRD1[WT] and TLNRD1[2T] could bind actin similarly, TLNRD1[2E] could not (Fig. 7 D). These findings concord with the hypothesis that the actin and CCM2 binding sites on TLNRD1 might overlap, rendering each TLNRD1 monomer incapable of binding to both actin and CCM2 simultaneously. To investigate this, we performed actin-binding assays in the presence or absence of CCM2[CTH]. These experiments revealed that TLNRD1[WT] lost its ability to bind to actin in the presence of CCM2[CTH] (Fig. 7 D). Importantly, TLNRD1[2T] maintained its ability to bind actin even in the presence of CCM2[CTH], although the efficiency was somewhat reduced (Fig. 7 D). Our results demonstrate that the TLNRD1–CCM2 and the TLNRD1–actin interactions are mutually exclusive. To assess the functional relevance of this, we next investigated in vitro if CCM2[CTH] would inhibit the ability of TLNRD1 to bundle actin (Fig. 7 E). These experiments revealed that while TLNRD1[WT] bundles actin effectively, as we reported previously (Cowell et al., 2021), the addition of CCM2[CTH] inhibited TLNRD1[WT] actin bundling (Fig. 7 E). In contrast, TLNRD1[2E] was unable to bundle actin (Fig. 7 E). These results indicate that CCM2 modulates TLNRD1 localization as well as actin binding and bundling activity.

### The TLNRD1–CCM2 interaction modulates actin stress fibers and focal adhesion formation

To further investigate the role of the TLNRD1–CCM2 interaction in endothelial cells, we focused on the organization of the actin cytoskeleton and focal adhesion formation. TLNRD1-silenced cells were transfected with various constructs, including GFP as a control, TLNRD1 wild type to evaluate functional restoration, TLNRD1[2T] to assess the impact of disrupting the TLNRD1–CCM2 interaction, and TLNRD1[F250D] to investigate the effects of blocking TLNRD1's dimerization capabilities. Indeed, we previously reported that TLNRD1[F250D] renders TLNRD1 monomeric, suppressing its actin-bundling activity (Cowell et al., 2021). Cells were also cotransfected with lifeact-RFP to enable a detailed visualization of the actin cytoskeleton in individual cells within the monolayer. Paxillin staining was employed to visualize focal adhesions (Fig. 8, A and B).

In these experiments, TLNRD1 silencing led to a noticeable increase in cell area, a higher number of stress fibers, a higher coverage of cells by paxillin-positive adhesions, and a marked reduction in filopodia formation (Fig. 8, A and B). Significantly, the re-expression of TLNRD1-WT in siTLNRD1 cells reversed these phenotypes (Fig. 8, A and B). Conversely, the re-expression of the TLNRD1[F250D] mutant did not, which demonstrates the importance of TLNRD1's dimerization for TLNRD1 functions.

Interestingly, the re-expression of the TLNRD1[2T] mutant in TLNRD1-depleted cells led to a mixed phenotype. It fully restored filopodia formation but only partially restored phenotypes related to the enhanced cell area, stress fiber formation, and paxillin-positive adhesion coverage (Fig. 8, A and B). These results indicate that while a mutant incapable of binding to CCM2 can compensate for certain TLNRD1 functions, the full spectrum of TLNRD1's role in cytoskeletal and adhesion regulation requires an intact TLNRD1–CCM2 interaction.

### Discussion

Here we report that TLNRD1 is expressed in the vasculature in vivo and that silencing of TLNRD1 expression results in vascular phenotypes both in vitro and in vivo. We demonstrate that TLNRD1 is a member of the CCM complex and that TLNRD1 interacts directly and with high affinity to CCM2. We also show that the TLNRD1–CCM2 binding interface involves a hydrophobic groove on TLNRD1 and hydrophobic residues of CCM2. Our findings indicate that CCM2 controls TLNRD1 localization to the cytoplasm and inhibits its actin-bundling activity and that the CCM2–TLNRD1 interaction impacts endothelial actin stress fiber and focal adhesion formation. As TLNRD1 expression is altered in CCM lesions, we propose that TLNRD1 could be a novel actor in the CCM disease.

We find that TLNRD1 is a member of the CCM complex and that TLNRD1 depletion leads to vascular phenotypes both in vitro and in vivo. The CCM complex is best known for modulating and maintaining the vasculature, and loss of function mutations of CCM complex components leads to CCM (Fischer et al., 2013). Our results are consistent with work from others, as a recent paper from Schnitzler et al. identified TLNRD1 to be linked to the CCM pathway (Schnitzler et al., 2024). Schnitzler et al. also reported that TLNRD1 targeting affects endothelial cell permeability and zebrafish heart development (Schnitzler et al., 2024). Interestingly, our results show that TLNRD1 silencing initially delays the formation of an

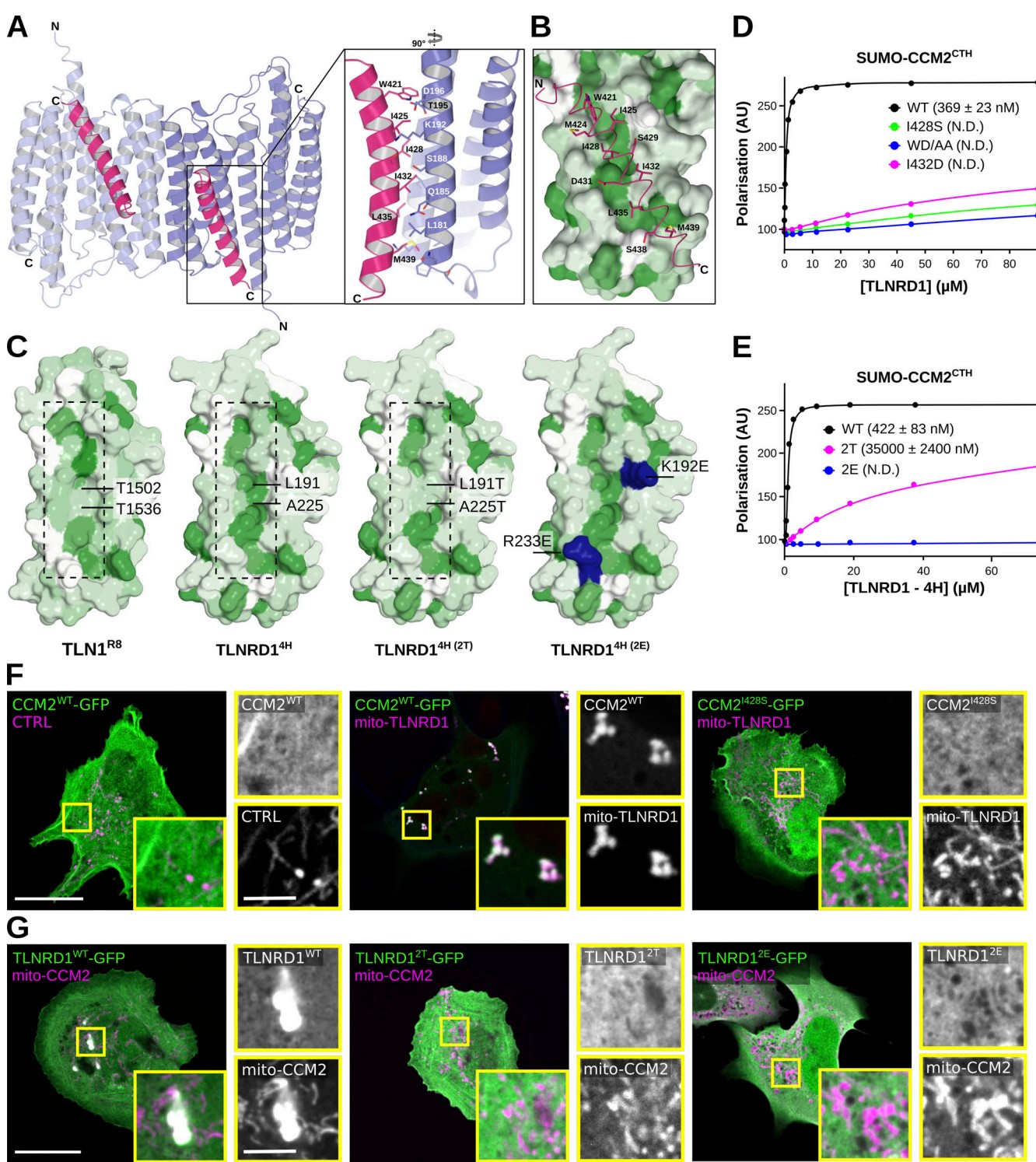

Figure 5. **The TLNRD1–CCM2 binding interface involves a hydrophobic groove on TLNRD1 and hydrophobic residues of CCM2. (A and B)** Modeling of the TLNRD1–CCM2 complex using ColabFold using the TLNRD1 crystal structure (PDB accession no. 6XZ4) as a template. **(A)** Overall view of the predicted complex. The TLNRD1 monomers are colored blue, and the CCM2$^{CTH}$ helices are colored pink. The TLNRD1–CCM2 binding area is magnified, and the residues contributing to the interface are shown as sticks. **(B)** TLNRD1 has a hydrophobic channel (green) on the surface, which could facilitate CCM2 (pink) binding. The TLNRD1 four-helix module was colored by hydrophobicity using the AA index database (entry FASG890101 [Nakai et al., 1988]) in PyMOL, where green denotes hydrophobic residues and white polar residues. CCM2$^{CTH}$ is shown as sticks and predominantly contacts the hydrophobic region on TLNRD1$^{4H}$. **(C)** Comparison of the hydrophobic channel on the surface of TLNRD1$^{4H}$ and the equivalent region on TLN1$^{R8}$. The TLNRD1$^{2T}$ mutant was designed to mimic the surface of TLN1$^{R8}$. The green color denotes hydrophobic residues. On the TLNRD1$^{2E}$, the mutated basic residues are highlighted in blue. **(D)** Fluorescence polarization was used to determine the $K_d$ of the interaction between TLNRD1 and various SUMO-CCM2$^{CTH}$ constructs (WT, I428S, I432D, and W412A/D422A). $K_d$ values (nM) are shown in parentheses. ND, not determined. **(E)** Fluorescence polarization was used to determine the $K_d$ of the interaction between CCM2$^{CTH}$ and various TLNRD1$^{4H}$ constructs (WT, 2T, and 2E). $K_d$ values (nM) are shown in parentheses. ND, not determined. **(F)** U2OS cells expressing various GFP-tagged CCM2

constructs and mito-TLNRD1-mScarlet or mito-mScarlet (CTRL) were imaged using a spinning disk confocal microscope. Representative single Z-planes are displayed. See also Fig. S5, E and F. Scale bars: (main) 25 µm and (inset) 5 µm. **(G)** U2OS cells expressing various GFP-tagged TLNRD1 constructs and mito-CCM2-mScarlet or mito-mScarlet (CTRL) were imaged using a spinning disk confocal microscope. Representative maximum intensity projections are displayed. Scale bars: (main) 25 µm and (inset) 5 µm.

---

impermeable monolayer, yet it subsequently enhances recovery of barrier function following thrombin-induced damage. At this point, the dual roles of TLNRD1 remain elusive; however, we speculate that its complex functions may derive from its dynamic involvement in modulating the actin cytoskeleton and its contributions to the CCM complex. Future studies will focus on dissecting TLNRD1's roles during various endothelial barrier formation and function stages.

75% of familial CCM cases are attributed to mutations in KRIT, CCM2, and PDCD10 (Chohan et al., 2019). To our knowledge, TLNRD1 mutations have not yet been reported in CCM patients, and future work will aim at sequencing TLNRD1 in familial CCM samples. Instead, a reanalysis of previous work indicated that TLNRD1 expression is upregulated in CCM lesions (Subhash et al., 2019). Interestingly, TLNRD1 expression is also upregulated in KRIT1 but not in PDCD10 knock-out mice (Koskimäki et al., 2019b, 2019a). TLNRD1 is also overexpressed in patients with dilated cardiomyopathy and ischemic heart disease (Liu et al., 2015) and is associated with significant stroke risk (Mishra et al., 2022). Taken together, TLNRD1 is emerging as an important regulator of endothelial cell function in vitro and in vivo that is misregulated in vascular diseases.

We also found that TLNRD1 interacts with the CCM complex via CCM2. Our results are consistent with previous work as TLNRD1 was found to coprecipitate with CCM2 (Schnitzler et al., 2024), and TLNRD1 and CCM2 scored in a yeast two-hybrid screen (Luck et al., 2020). We identified that the TLNRD1–CCM2 interaction involves the C-terminal helix in CCM2 that interacts with TLNRD1's four-helix domain. Interestingly, this interaction is primarily hydrophobic, explaining why TLN1 R7R8 does not bind to CCM2 despite being structurally homologous to TLNRD1 (Cowell et al., 2021). Importantly, CCM2 binds to TLNRD1 on the same surface as actin binds, and we found that the TLNRD1–actin and TLNRD1–CCM2 interactions are mutually exclusive. TLNRD1 interacts with CCM2 with high affinity, and our results indicate that CCM2 inhibits TLNRD1 actin-bundling activity. As the affinity of the TLNRD1–actin interaction is close to that of the TLNRD1–CCM2 interaction, an attractive hypothesis is that local actin dynamics could regulate the TLNRD1–CCM2 and the TLNRD1–actin interactions.

One key CCM complex function within cells is to restrain cellular contractility, thereby limiting the formation of stress fibers that compromise endothelial barrier function. For instance, previous work reported that CCM2 deletion increases actin stress fiber formation in endothelial cells (Fischer et al., 2013). Notably, each component of the CCM complex inhibits the RhoA-ROCK (Rho-associated coiled coil-forming kinase) pathway, subsequently preventing myosin light chain (MLC) phosphorylation (Riolo et al., 2021; Su and Calderwood, 2020; Stockton et al., 2010; Whitehead et al., 2009; Hartmann et al., 2015; Lisowska et al., 2018; Zheng et al., 2010; Crose et al., 2009).

In this study, we report that silencing TLNRD1 leads to significant alterations in endothelial cell morphology and cytoskeletal architecture, evidenced by an expanded cell area, an increase in the number of stress fibers, enhanced coverage by paxillin-positive adhesions, and a substantial decrease in filopodia formation, while not affecting MLC phosphorylation levels. Notably, our findings highlight the critical role of the TLNRD1–CCM2 interaction in the modulation of actin stress fiber and focal adhesion formation in endothelial cells. The necessity of an intact dimerization motif in TLNRD1 for these phenotypes, coupled with the observation that TLNRD1 interactions with actin and CCM2 are mutually exclusive, suggests that TLNRD1's cellular roles are likely context-dependent. We speculate that in environments rich in F-actin, TLNRD1 preferentially binds and bundles F-actin, facilitating processes such as filopodia formation. Conversely, we speculate that in the presence of CCM2, TLNRD1 pivots toward supporting the CCM complex's function, potentially aiding in the formation of CCM complex multimers and the assembly of signaling platforms. Our data also do not exclude the intriguing possibility that TLNRD1 may act as a molecular bridge, linking the CCM complex with the actin cytoskeleton. Future investigations will be aimed at further deciphering TLNRD1's contributions to CCM complex assembly.

## Materials and methods

### Cells
U2OS osteosarcoma cells and HEK293 (human embryonic kidney) cells were grown in DMEM (Dulbecco's Modified Eagle's Medium; D1152; Sigma-Aldrich) supplemented with 10% fetal bovine serum (FBS) (S1860; Biowest). U2OS cells were purchased from DSMZ (Leibniz Institute DSMZ-German Collection of Microorganisms and Cell Cultures, Braunschweig DE, ACC 785). HEK293 cells were provided by ATCC (CRL-1573). Human Umbilical Vein Endothelial Cells (HUVEC) (PromoCell C-12203) were grown in Endothelial cell growth medium (ECGM) (PromoCell C-22010) supplemented with supplemental mix (Promocell C-39215) and 1% penicillin–streptomycin (Sigma-Aldrich). Endothelial primary cells from P0 (commercial vial) were expanded to a P3 stock stored in a –80°C freezer to standardize the experimental replicates.

### Antibodies and reagents
The anti-TLNRD1 antibody used for western blot (1:1,000) was raised in rabbits against recombinantly expressed human TLNRD1 (residues 1–362) by Capra Science. The rabbit anti-TLNRD1 used to stain the brain section was purchased from Atlas Antibodies (1:200 for IF, HPA071716). Other rabbit antibodies used in this study include anti-KRIT1 (1:1,000 for WB, ab196025; Abcam), anti-fibronectin (1:200 for IF, f3648; Sigma-

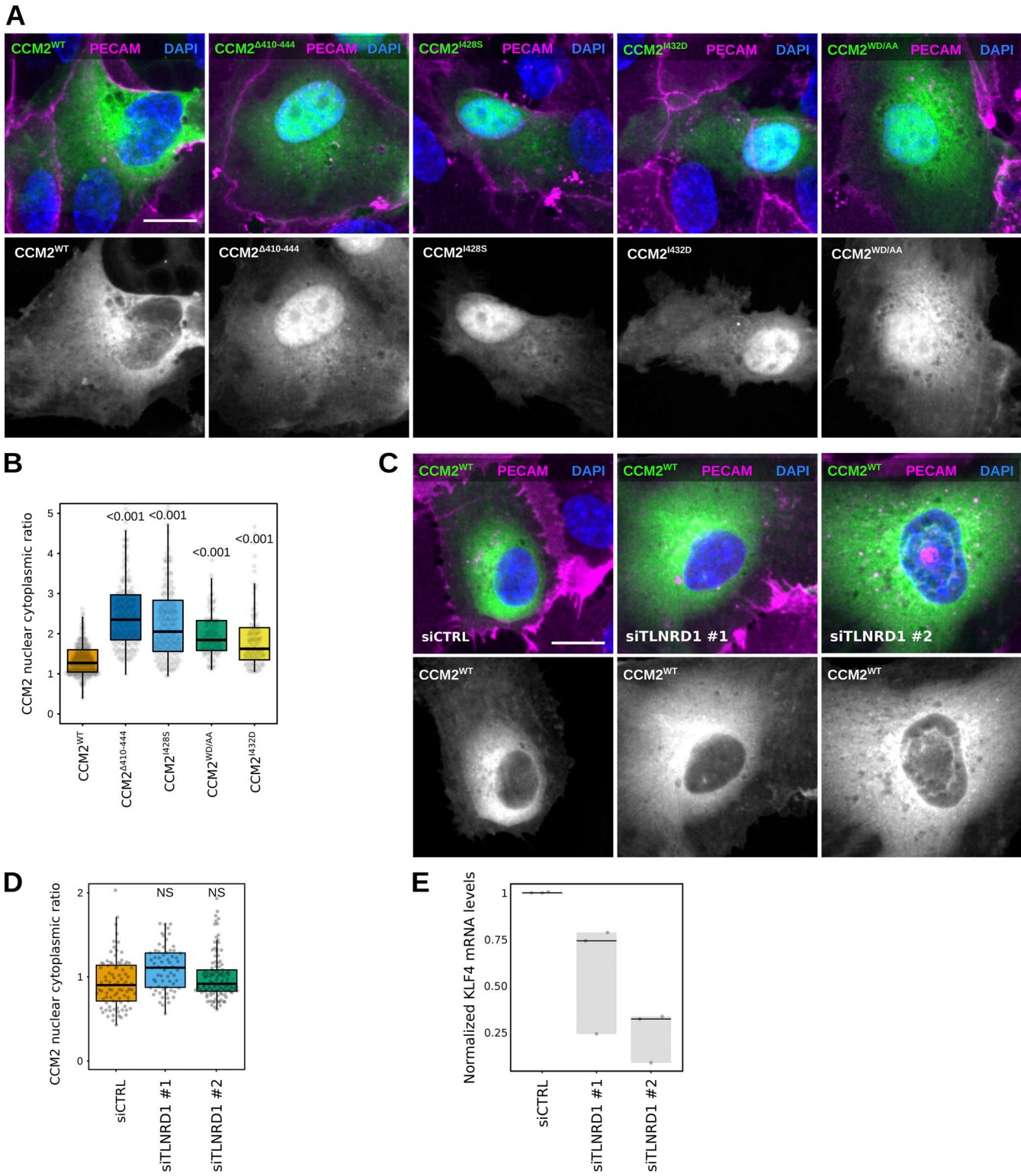

Figure 6. **TLNRD1 silencing does not impact CCM2 localization. (A and B)** HUVECs expressing various CCM2-GFP constructs were stained for DAPI and PECAM and imaged using a spinning disk confocal microscope. **(A)** SUM projections are displayed. Scale bar: 10 μm. **(B)** For each condition, the CCM2 nuclear-cytoplasmic ratio was quantified (three biological repeats, $n > 110$ cells per condition). **(C–E)** TLNRD1 expression was silenced in HUVECs using two independent siRNA. **(C and D)** Cells were then transfected to express CCM2-GFP and allowed to form a monolayer. Cells were fixed and stained for DAPI and PECAM and imaged using a spinning disk confocal microscope. **(C)** SUM projections are displayed. Scale bar: 10 μm. **(D)** For each condition, the CCM2 nuclear-cytoplasmic ratio was quantified (three biological repeats, $n > 60$ cells per condition). **(E)** HUVECs were then allowed to form a monolayer without flow stimulation. KLF4 expression levels were measured by qPCR. **(B and D)** The results are displayed as Tukey boxplots. The whiskers (shown here as vertical lines) extend to data points no further from the box than 1.5× the interquartile range. For all panels, the P values were determined using a randomization test. NS indicates no statistical difference between the mean values of the highlighted condition and the control.

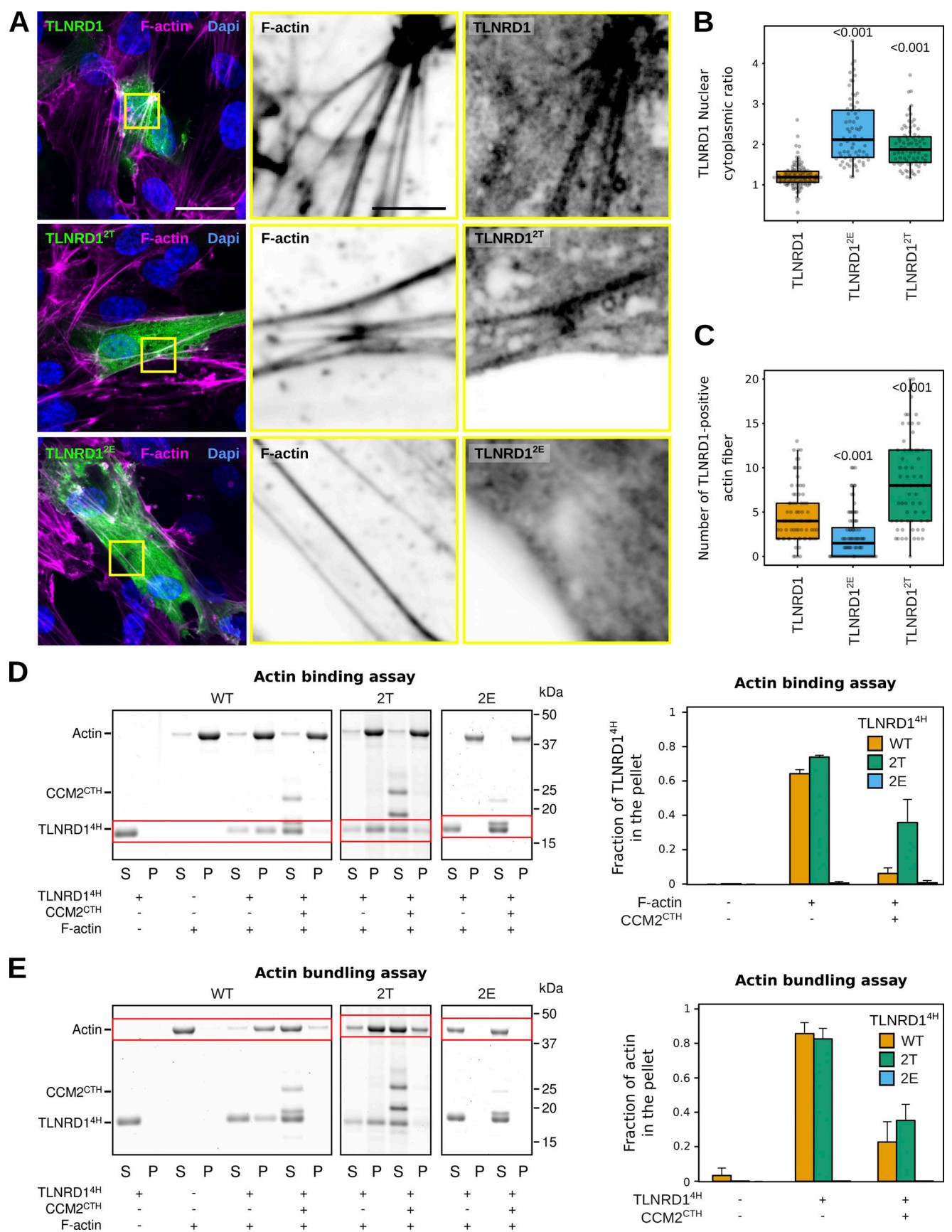

Figure 7. **CCM2 modulates TLNRD1 localization and bundling activity. (A–C)** HUVECs expressing various TLNRD1-GFP constructs were stained for DAPI and F-actin and imaged using a spinning disk confocal microscope. **(A)** Single Z-planes are displayed. The yellow squares highlight magnified ROIs. Scale bars:

(main) 25 μm and (inset) 5 μm. **(B and C)** For each condition, the TLNRD1 nuclear-cytoplasmic ratio and the number of TLNRD1-positive actin fibers were quantified (see methods for details), and results are displayed as Tukey boxplots (three biological repeats, *n* > 72 cells per condition). The P values were determined using a randomization test. **(D and E)** Actin co-sedimentation assay with various TLNRD1$^{4H}$ mutants in the presence or absence of CCM2$^{CTH}$. Centrifugation at high (D, 48,000 rpm) or low (E, 16,000 rpm) speeds can distinguish between F-actin binding and bundling capability. Representative SDS-PAGE gels are displayed. The quantification was performed using densitometry, and the fraction of TLNRD1 (D) and F-actin (E) present in the pellet was plotted. Standard deviation from three independent repeats are represented as error bars. Source data are available for this figure: SourceData F7.

Aldrich), anti-GFP (1:1,000 for WB, A11121; LifeTechnologies), and anti-Myosin light chain (phospho S20) antibody (1:200 for IF, Abcam, ab2480). Mouse antibodies were anti-CCM2 (1:1,000 for WB, MA5-25668; Thermo Fisher Scientific), anti-PECAM (1:200 for IF, 37-0700; Invitrogen), anti-Paxillin (1:200 for IF, ab32084; Abcam), and anti-GAPDH (1:1,000 for WB, 5G4; Hytest).

### Plasmids used for cell studies
The pMTS_mScarlet-i_N1 (mito-mScarlet) was a gift from Dorus Gadella (University of Amsterdam, Amsterdam, Netherlands; plasmid #85059; Addgene; RRID: Addgene_85059) (Bindels et al., 2017). mRuby-Lifeact-7 was a gift from Michael Davidson (plasmid #54560; Addgene; RRID:Addgene_54560). mScarlet-MYO10 was described previously and is available on Addgene (plasmid #145179; Addgene; RRID: Addgene_145179) (Jacquemet et al., 2019). The GFP-TLNRD1 was as described previously (Gingras et al., 2010), and the GFP-TLNRD1-4H was generated from it. The GFP-TLNRD1 and GFP-TLNRD1-4H are in the vectors pEGFP-C1 and pEGFP-N3 respectively and will be deposited in Addgene (https://www.addgene.org/Ben_Goult/).

The ITG1BP1-GFP construct (ICAP-1) was a gift from Daniel Bouvard (University of Grenoble, Grenoble, France). The PDCD10-mEmerald (PDCD10-GFP) and the CCM2-mCherry constructs were generated by the Genome Biology Unit core facility cloning service (Research Programs Unit, HiLIFE Helsinki Institute of Life Science, Faculty of Medicine, University of Helsinki, Biocenter Finland) by transferring the PDCD10 entry clone (100005213) into pcDNA6.2/N-emGFP-DEST and the CCM2 entry clone (100073308) into pcDNA6.2/C-mCherry-DEST using a standard LR reaction protocol.

The mito-CCM2-mScarlet-I (mito-CCM2) and mito-PDCD10-mScarlet-I (mito-PDCD10) constructs were created by inserting a custom gene block (IDT) in the pMTS_mScarlet-i_N1 plasmid (#85059; Addgene) using the XhoI/EcoRI sites. The gene blocks used are available in Table S2.

The TLNRD1_5H_eGFP (TLNRD1$^{5H}$), CCM2_wt_eGFP (CCM2$^{WT}$), CCM2_d410-444_eGFP (CCM2$^{\Delta CTH}$), CCM2_I428-S_eGFP (CCM2$^{I428S}$), CCM2_I432D_eGFP (CCM2$^{I432D}$), CC-M2_W421AD422A_eGFP (CCM2$^{WD/AA}$), TLNRD1_2T_eGFP (TLNRD1$^{2T}$), TLNRD1_2E_eGFP (TLNRD1$^{2E}$), and mito-TLNRD1-mScarlet-I constructs were purchased from GenScript. Briefly, the gene fragments were synthesized using gene synthesis and cloned into pcDNA3.1(+)-N-eGFP using the BamHI/XhoI sites. The full plasmid sequences are available in Table S2. All constructs were sequence verified. These plasmids will be available on Addgene (https://www.addgene.org/Guillaume_Jacquemet/).

### Plasmids used for producing recombinant proteins
The FL and 4H TLNRD1 were described previously (Cowell et al., 2021) and are available on Addgene (plasmids #159384 and #159386; Addgene). The FL CCM2 constructs were purchased from GeneArt and subcloned into pET151. GST-CCM2 constructs were produced by subcloning the CCM2 constructs into the XmaI/SacI sites of pET49b using ligation-independent cloning. The HisSUMO-tagged CCM2(413-438) was subcloned into a modified pET47b vector encoding a hexahistidine tag fused to SUMO (where all lysine residues were mutated to arginine residues) followed by an HRV-3C cleavage site (provided by Dr. I.A. Taylor, The Francis Crick Institute, London, UK) using ligation-independent cloning. Mutations were introduced by site-directed mutagenesis. All constructs were sequence verified. The recombinant expression vectors of GST-CCM2(FL), HisSumo-CCM2(413-438), TLNRD1(4H)-2T, and TLNRD1(4H)-2E will be deposited in Addgene (https://www.addgene.org/Ben_Goult/).

### Cell transfection
U2OS cells were transfected using Lipofectamine 3000 and the P3000 Enhancer Reagent (L3000001; Thermo Fisher Scientific) according to the manufacturer's instructions. HUVECs were transfected using the Neon Transfection System (MPK5000; Thermo Fisher Scientific) and the Neon Transfection System 10 μl Kit (MPK1025; Thermo Fisher Scientific) according to the manufacturer's instructions. Briefly, 150 k cells and 1 μg of plasmid DNA were used for each transfection (1.350 pulse voltage and 30 ms pulse width). Transfected cells were seeded in a glass bottom μ-slide eight-well (80807; Ibidi) with a glass bottom precoated with warm ECGM without antibiotics. 50 k untransfected cells/well were added after transfection. Cells were then grown for 48 h and fixed using prewarmed paraformaldehyde 4% in PBS (28908; Thermo Fisher Scientific) for 10 min at 37°C.

HEK293T cells were transfected using 100× polyethyleneimine reagent (Sigma-Aldrich). Cells were seeded in a 15-cm dish and allowed to reach 80% confluence before transfection. The transfection mixture consisted of 8 μg plasmids diluted in 500 ml OptiMEM and 54 μl of 1× polyethyleneimine (diluted with 150 mM NaCl) preincubated with 446 ml of Opti-MEM for 5 min at room temperature. Transfection mixtures were incubated for 20 min at room temperature. The medium was then removed from the cells and replaced with 10 ml growth medium and the transfection mixture. Following 10 h of incubation, the transfection mixture was removed, a fresh medium was added, and transfected cells were used the following day.

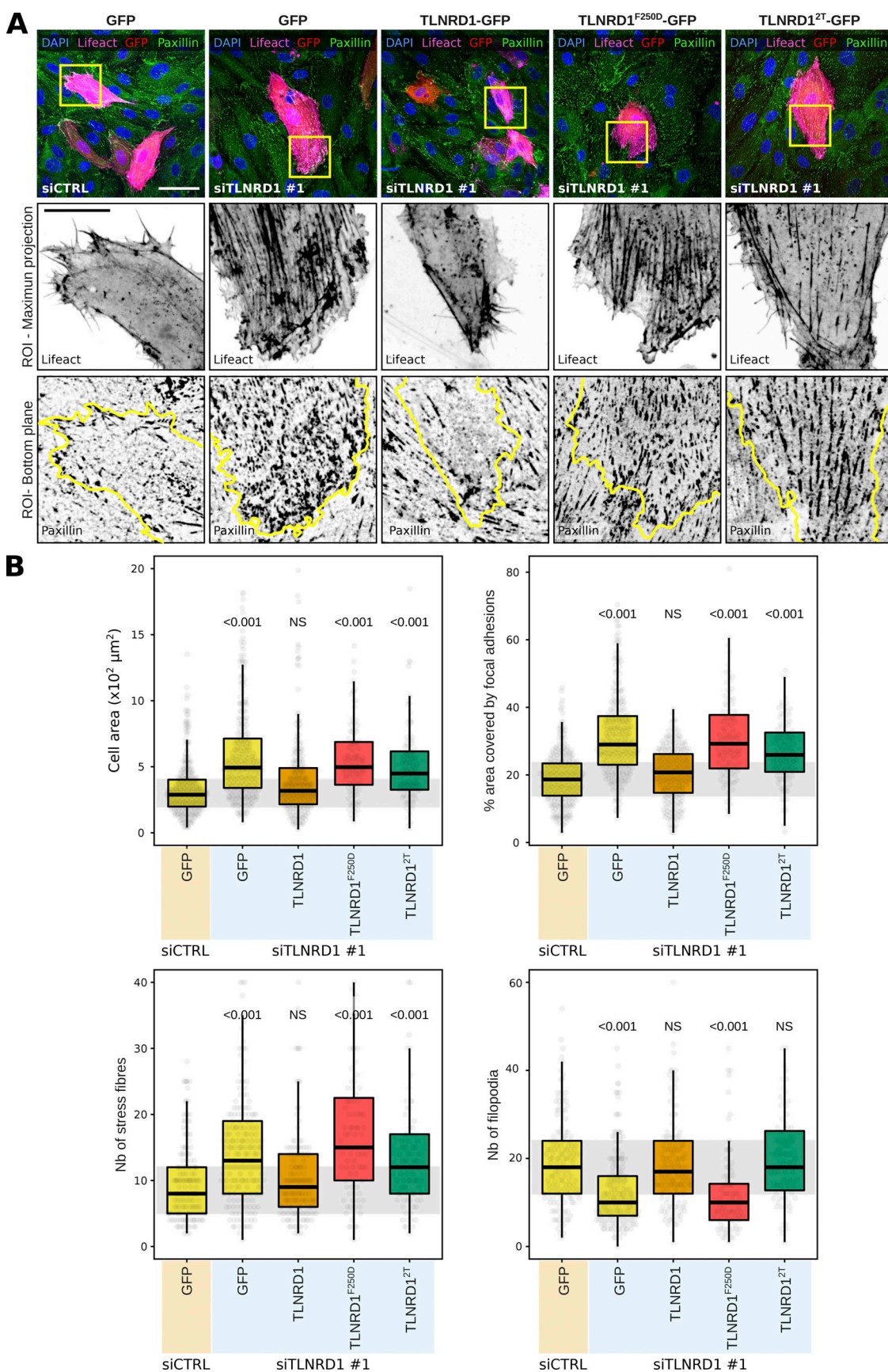

Figure 8. **CCM2 modulates TLNRD1 localization and bundling activity. (A and B)** HUVECs treated with siCTRL or siTLNRD1 siRNA and expressing lifeact-RFP along with GFP, TLNRD1-GFP, TLNRD1$^{2T}$-GFP, or TLNRD1$^{F250D}$-GFP were fixed and stained for DAPI and paxillin, followed by imaging using a spinning disk

confely microscope. **(A)** Maximal intensity projections of representative fields of view. Highlighted within yellow squares are ROIs selected for magnification. The upper ROI panel presents maximal intensity projections showcasing lifeact-RFP and GFP-positive cells. In contrast, the lower ROI panels concentrate on the basal plane to showcase the paxillin-positive adhesions, with yellow outlines delineating the contours of GFP-positive cells. Scale bars: (main) 50 µm and (inset) 20 µm. **(B)** Quantitative analysis was performed on GFP-positive cells, evaluating various parameters: cell area, the proportion of the cell area covered by paxillin-positive adhesions, and the number of actin stress fibers and filopodia (four biological repeats, >60 fields of view per condition). The results are shown as Tukey boxplots. The whiskers (shown here as vertical lines) extend to data points no further from the box than 1.5× the interquartile range. The P values were determined using a randomization test.

## siRNA-mediated gene silencing

The expression of KRIT1 was suppressed in U2OS cells using 83 nM siRNA and Lipofectamine 3000 (L3000001; Thermo Fisher Scientific) according to the manufacturer's instructions. The expression of TLNRD1 was suppressed in HUVECs using 50 nM siRNA and Lipofectamine RNAiMAX (13778075; Thermo Fisher Scientific) following the manufacturer's instructions. HUVECs were seeded on fibronectin-coated wells or microchannels in the siRNA-containing solution for 2 h. Then, normal media was added to the cells. The following day, the transfection media was replaced. These additional steps maximized siRNA entry in the cells while preserving cell viability. Cell experiments were performed after 72 h of treatment with siRNA in all cases. siRNAs used were AllStars Negative siRNA (SI03650318), TLNRD1 siRNA #1 (SI04314569), TLNRD1 siRNA #2 (SI04362820), KRIT1 siRNA #1 (SI02777173), and KRIT1 siRNA #2 (SI03054499), all from Qiagen.

## Primers and qPCR

RNAs were extracted and purified using an RNeasy Mini Kit (74104; Qiagen) according to the manufacturer's instructions, including a DNase digestion using an RNase-Free DNase (79254; Qiagen). The RNA concentration and quality were assessed using a Nanodrop 2000 (Thermo Fisher Scientific). cDNAs were then synthesized using iScript cDNA (1708890; Bio-rad) according to the manufacturer's instructions. Real-time quantitative PCR (RT-qPCR) was performed using QuantStudio 3 (A28567; Thermo Fisher Scientiifc) with PowerUp SYBR Green Mastermix (A25741; Life Technologies). The reaction mixture consisted of 4 ng of cDNA, primers, and master mix and was run under the following conditions: hold (50°C for 2 min, 95°C for 2 min), PCR (95°C for 15 s, 60°C for 1 min), and melt curve (95°C for 15 s, 60°C for 1 min and 95°C for 15 s). The expression levels of the target genes were normalized to the expression level of GAPDH using the [ΔΔCt method]. All experiments were performed in triplicate, and the results were analyzed using QuantStudio Design & Analysis Software 2.6.0.

## SDS-PAGE and quantitative Western blotting

Protein extracts were separated under denaturing conditions by SDS-PAGE and transferred to nitrocellulose membranes using a Mini Blot Module (B1000; Invitrogen). Membranes were blocked for 30 min at room temperature using 1× StartingBlock buffer (37578; Thermo Fisher Scientific). After blocking, membranes were incubated overnight with the appropriate primary antibody (1:1,000 in blocking buffer), washed three times in PBS, and probed for 1 h using a fluorophore-conjugated secondary antibody diluted 1:5,000 in the blocking buffer. Membranes were washed thrice using PBS over 30 min and scanned using an iBright FL1500 imaging system (Invitrogen).

## GFP-trap pulldown

Cells transiently expressing bait GFP-tagged proteins were lysed in a buffer containing 20 mM HEPES, 75 mM NaCl, 2 mM EDTA, 1% NP-40, as well as a cOmplete protease inhibitor tablet (5056489001; Roche), and a phosphatase inhibitor mix (Roche, 04906837001). Lysates were then centrifuged at 13,000 $g$ for 10 min at 4°C. Clarified lysates were incubated with GFP-Trap agarose beads (#gta-20; Chromotek) overnight at 4°C. Complexes bound to the beads were isolated by centrifugation, washed three times with ice-cold lysis buffer, and eluted in Laemmli reducing sample buffer.

## Mass spectrometry analysis

Affinity-captured proteins were separated by SDS-PAGE and allowed to migrate 10 mm into a 4–12% polyacrylamide gel. Following staining with InstantBlue (Expedeon), gel lanes were sliced into five 2-mm bands. The slices were washed using a solution of 50% 100 mM ammonium bicarbonate and 50% acetonitrile until all blue colors vanished. Gel slices were washed with 100% acetonitrile for 5–10 min and then rehydrated in a reducing buffer containing 20 mM dithiothreitol in 100 mM ammonium bicarbonate for 30 min at 56°C. Proteins in gel pieces were then alkylated by washing the slices with 100% acetonitrile for 5–10 min and rehydrated using an alkylating buffer of 55 mM iodoacetamide in 100 mM ammonium bicarbonate solution (covered from light, 20 min). Finally, gel pieces were washed with 100% acetonitrile, followed by washes with 100 µl 100 mM ammonium bicarbonate, after which slices were dehydrated using 100% acetonitrile and fully dried using a vacuum centrifuge. Trypsin (0.01 µg/µl) was used to digest the proteins (37°C overnight). After trypsinization, an equal amount of 100% acetonitrile was added, and gel pieces were further incubated at 37°C for 15 min, followed by peptide extraction using a buffer of 50% acetonitrile and 5% formic acid. The buffer with peptides was collected, and the sample was dried using a vacuum centrifuge. Dried peptides were stored at –20°C. Before LC-ESI-MS/MS analysis, dried peptides were dissolved in 0.1% formic acid.

The LC-ESI-MS/MS analyses were performed on a nanoflow HPLC system (Easy-nLC1000; Thermo Fisher Scientific) coupled to the Q Exactive mass spectrometer (Thermo Fisher Scientific) equipped with a nano-electrospray ionization source. Peptides were first loaded on a trapping column and subsequently separated inline on a 15-cm C18 column (75 µm × 15 cm, ReproSil-Pur 5 µm 200 Å C18-AQ, Dr. Maisch HPLC GmbH, Ammerbuch-Entringen, Germany). The mobile phase consisted of water with 0.1% formic acid (solvent A) and acetonitrile/water (80:20 [vol/vol]) with 0.1% formic acid (solvent B). A linear 30-min gradient from 6% to 39% was used to elute peptides. MS data

was acquired automatically by using Thermo Xcalibur 3.0 software (Thermo Fisher Scientific). An information-dependent acquisition method consisted of an Orbitrap MS survey scan of mass range 300–2,000 m/z followed by HCD fragmentation for the 10 most intense peptide ions.

Raw data from the mass spectrometer were submitted to the Mascot search engine using Proteome Discoverer 1.5 (Thermo Fisher Scientific). The search was performed against the human database SwissProt_2018_04, assuming the digestion enzyme trypsin, a maximum of two missed cleavages, an initial mass tolerance of 10 ppm (parts per million) for precursor ions, and a fragment ion mass tolerance of 0.020 Da. Cysteine carbamidomethylation was set as a fixed modification, and methionine oxidation was set as a variable modification.

Three biological replicates were combined to generate the TLNRD1 dataset. Proteins enriched at least threefold in TLNRD1-GFP over GFP (based on normalized spectral count) and detected with at least six spectral counts (across all repeats) were considered putative TLNRD1 binders.

The fold-change enrichment and the significance of the association used to generate the volcano Plot (Fig. 1 B) were calculated in Scaffold 5 (version 5.2.0; Proteome Software, Inc.) using a Fisher's exact $t$ test corrected using the Benjamini-Hochberg method. To generate the protein–protein interaction network displayed in Fig. 1 C, enriched proteins were mapped onto a merged human interactome consisting of PPIs reported in STRING (v. 11.5) (Szklarczyk et al., 2019) directly in Cytoscape (version 3.9.1) (Shannon et al., 2003).

## Zebrafish experiments

The adult zebrafish of kdrl:mCherry-CAAX(s916) strain (Hogan et al., 2009) were housed in an Aqua Schwarz stand-alone rack (Aqua Schwarz GmbH). The husbandry of adult fish was carried out under license no. MMM/465/712-93 (issued by the Finnish Ministry of Forestry and Agriculture). Embryos were obtained via natural spawning in breeding tanks.

Single-guide RNAs (sgRNA) targeting the tlnrd1 locus were designed using CHOPCOP software (Labun et al., 2019). SgRNAs were synthesized using an sgRNA synthesis kit (New England Biolabs) as described by the manufacturer and purified using RNA-clean 25 columns (ZYMO Research) and using the following oligonucleotides designed to target tlnrd1: tlnrd1-sgRNA#9 (5′-TTCTAATACGACTCACTATAGCTCGGGGAAATCAGATAG CGGTTTTAGAGCTAGA-3′), tlnrd1-sgRNA#14 (5′-TTCTAATAC GACTCACTATAGTTAGCGGCAGCTTGCAACAAGTTTTAG AGCTAGA-3′), tlnrd1-sgRNA#24 (5′-TTCTAATACGACTCACTA TAGCTATGGCTAGTAGTGGCTCGGTTTTAGAGCTAGA-3′), and tlnrd1-sgRNA#25 (5′-TTCTAATACGACTCACTATAGCACACTAC TATGGCTAGTAGGTTTTAGAGCTAGA-3′). As a control, sgRNAs targeting the slc45a2 locus were used (Heliste et al., 2020). The slc45a2 gene encodes a transporter protein that is crucial in melanin synthesis. In zebrafish, the disruption of the slc45a2 function leads to a visible loss of pigmentation in the eyes, an easily observable and quantifiable phenotype. We utilized slc45a2 as an additional control in our experiments because it is a reliable positive control for CRISPR-mediated effects that is often used (Heliste et al., 2020). The loss of pigmentation in

embryos where slc45a2 is disrupted is a clear indicator of successful genome editing. Purified sgRNAs targeting tlnrd1 and slc45a2 were pooled separately and complexed with recombinant Cas9 protein (New England Biolabs) in vitro using 300 mM KCl buffer for 5 min at +37°C. The solution was injected into one- to four-cell stage zebrafish embryos. The efficacy of tlnrd1 targeting was confirmed by extracting DNA as described earlier (Meeker et al., 2007) and amplifying targeted regions using nested-PCR (tlnrd1-L3 [5′-TCATTTACATGGCACGAAGAAC-3′], tlnrd1-R4 [5′-GGTGAGGTTCTTCAGGATGTTC-3′], tlnrd1-L4 [5′-C GAGTGAAGTTTCATGTTTTCG-3′], tlnrd1-R3 [5′-ATAGACAGCTCCT TGGTTCTGG-3′]), and Sanger sequencing of amplicons. TIDE software (Brinkman et al., 2014) was used to determine the mutagenesis efficacy from sequencing chromatograms. Zebrafish embryos were incubated at 28.5°C in E3 medium (5 mM NaCl, 0.17 mM KCl, 0.33 mM CaCl$_2$, 0.33 mM MgSO$_4$) supplemented with 0.2 mM phenyl-thiourea until anesthetized and analyzed. Embryos were imaged using a Nikon Eclipse Ti2 widefield microscope equipped with a 2× objective or a Zeiss stereoLumar fluorescence stereomicroscope. Vascular measurements were manually measured using Fiji. Heart rate measurements were conducted using Fiji using kymographs (Schindelin et al., 2012).

## Light microscopy setup

The spinning-disk confocal microscope used was a Marianas spinning-disk imaging system with a Yokogawa CSU-W1 scanning unit on an inverted Zeiss Axio Observer Z1 microscope controlled by SlideBook 6 (Intelligent Imaging Innovations, Inc.). Images were acquired using either an Orca Flash 4 sCMOS camera (chip size 2,048 × 2,048; Hamamatsu Photonics) or an Evolve 512 EMC-CD camera (chip size 512 × 512; Photometrics). The objectives used were 40× (NA 1.1 water, Zeiss LD C-Apochromat) and 63× oil (NA 1.4 oil, Plan-Apochromat, M27) objectives.

The structured illumination microscope (SIM) used was DeltaVision OMX v4 (GE Healthcare Life Sciences) fitted with a 60× Plan-Apochromat objective lens, 1.42 NA (immersion oil RI of 1.516) used in SIM illumination mode (five phases x three rotations). Emitted light was collected on a front-illuminated pco.edge sCMOS (pixel size 6.5 μm, readout speed 95 MHz; PCO AG) controlled by SoftWorx.

The confocal microscope used was a laser scanning confocal microscope LSM880 (Zeiss) equipped with an Airyscan detector (Carl Zeiss) and a 40× water (NA 1.2) or 63× oil (NA 1.4) objective. The microscope was controlled using Zen Black (2.3), and the Airyscan was used in standard super-resolution mode.

## Quantification of endothelial monolayer integrity

We used a fibronectin accessibility assay to measure the integrity of the monolayer formed by HUVEC upon TLNRD1 silencing. SiRNA treatment was done while HUVECs were attached to glass-bottomed 24-well plates or microchannels (Ibidi μ-slide I LUER 0.4, 80177) previously coated with 10 μg/ml of fibronectin (341631; Sigma-Aldrich). In the flow-stimulated samples, the perfusion of the microchannels was done for 24 h (starting at 48 h post-siRNA treatment) with a perfusion speed of 400 μm/

sec (Follain et al., 2018; Osmani et al., 2021). At 72 h post-siRNA treatment, cells were fixed using prewarmed paraformaldehyde 4% in PBS (28908; Thermo Fisher Scientific) for 10 min at 37°C. When indicated, samples were stained directly so only the fibronectin localized underneath antibody-permeable junctions would be labeled. When samples were permeabilized to visualize all the deposited fibronectin, cells were incubated with a solution of 0.2% Triton X-100 in PBS (9002-93-1; Sigma-Aldrich) for 10 min at room temperature. Samples were then incubated with the primary antibodies diluted in PBS (1:200) for 1 h at room temperature. After three PBS washes, samples were incubated in the dark with the fluorescently conjugated secondary antibodies diluted in PBS (1:400) for 30 min at room temperature. Samples were imaged in 3D using a spinning disk confocal microscope, and the images were analyzed using Fiji. Briefly, after generating maximal intensity projections, the FN patches were automatically segmented using the "default" thresholding method and the "run analyze particles" function. The percentage of the field of view covered by fibronectin patches was then measured.

### Transendothelial electrical resistance measurements
TEER measurements were conducted using the xCELLigence Real-Time Cell Analyzer DP Instrument (W380; ACEA). HUVECs underwent gene silencing utilizing two distinct TLNRD1-targeting siRNAs or control siRNA (AllStars Negative Control siRNA) according to the previously described siRNA-mediated gene silencing protocol. Following a 24-h post-silencing period, 96-well E plates (300600910; Agilent) were coated with fibronectin (Cat. No. 341631; Sigma-Aldrich) at a concentration of 1 µg/ml for 45 min at 37°C. After a washing step with PBS, each well was filled with endothelial cell growth medium (ECGM), and an initial impedance measurement was conducted to establish a baseline. Cell seeding was performed by adding 20,000 HUVECs per well into three wells per experimental condition. TEER measurements were then systematically taken every 30 min over the course of 72 h. The culture medium was refreshed every 24 h to maintain optimal cell growth conditions. ECGM supplemented with thrombin (1 U/ml) was added to the cells 48 h after seeding for the thrombin treatment experiments. Impedance monitoring continued for an additional 24 h after the addition of thrombin to assess the effects of thrombin on endothelial barrier integrity.

In all experiments, the cell index, a unitless value provided by the xCELLigence Instrument, was used as a readout of the TEER. For the experiments displayed in Fig. 3 I, individual TEER trajectories were normalized to their final readings to study the establishment of the TEER over time. For the experiments displayed in Fig. S4 C, Iindividual TEER trajectories were normalized to the readings before the thrombin stimulation to study the effect of thrombin on TEER over time.

### Brain slice preparation
All animal experiments were ethically assessed and authorized by the National Animal Experiment Board and by following The Finnish Act on Animal Experimentation (Animal license number ESAVI/12558/2021). Mice (females Hsd: Athymic Nude-Foxn/1nu, Envigo) were housed in standard conditions (12-h light/dark cycle) with food and water available ad libitum. To collect brains, mice were sacrificed and placed in a ventral-down position. Incision at the base of the skull and cutting on both sides toward the sphenoid allowed the extraction of the brain in a single piece. Brains were rinsed in PBS and fixed in paraformaldehyde 4% (043368.9M; Thermo Fisher Scientific) at 4°C overnight.

Extracted brains were embedded in low melting point agarose (16520050; Thermo Fisher Scientific), and 100–200 µm thick brain slices were prepared using a vibratome (VT1200 S; Leica). Brain slices were stored in a 24-well plate and stained using a protocol adapted from (Fercoq et al., 2020). Briefly, sections were incubated with the permeabilization buffer (in PBS: 10% Horse Serum [vol/vol, 16050-122; Gibco], 1% BSA [wt/vol, 1003435812; Sigma-Aldrich], 0.3% TX-100 [vol/vol, 9002-93-1; Sigma-Aldrich] and 0.05% Sodium azide [wt/vol, 26628-22-8; Sigma-Aldrich]) for 1 h at room temperature before being rinsed once with the washing buffer (in PBS: 1% BSA, 0.1% TX-100 [vol/vol], and 0.05% sodium azide [wt/vol]). Samples were then incubated with primary antibodies, diluted in the antibody dilution buffer (in PBS: 10% horse serum [vol/vol], 1% BSA [wt/vol], 0.1% TX-100 [vol/vol], and 0.05% sodium azide [wt/vol]) on a rocker for 3 h at RT. Samples were washed thrice for 20 min with the washing buffer while on a rocker. Next, fluorescently conjugated secondary antibodies, diluted in the antibody dilution buffer, were added to the wells, and the plate was placed on a rocker for 3 h at RT. Samples were washed twice for 5 min with the washing buffer while on a rocker. Finally, the samples were rinsed for 5 min with PBS and then mounted on a slide using ProLong Glass Antifade Mountant (P36980; Thermo Fisher Scientific). Brain slices were then imaged using a spinning disk confocal microscope.

### Nuclear to cytoplasmic ratio measurements
HUVECs expressing the indicated CCM2 or TLNRD1 constructs were allowed to form a monolayer for 3 days before being fixed and stained for PECAM and DAPI. High-resolution imaging of the monolayers was conducted using a spinning disk confocal microscope. Subsequently, to quantify the nuclear-to-cytoplasmic ratio of TLNRD1 and CCM2, SUM projections of the acquired image stacks were generated using Fiji. Automatic cell and nuclei segmentation was performed using the ZeroCostDL4Mic cellpose notebook (Pachitariu and Stringer, 2022; von Chamier et al., 2021). Two separate models within Cellpose were employed—the "cyto2" model for segmenting the transfected cells and the "nuclei" model for segmenting the nuclei. Each segmentation was then manually validated within Fiji, following which cytoplasmic segmentation masks were created by excluding the nuclear mask from the overall cell mask. Finally, the average integrated fluorescence density within the nucleus was calculated and normalized against that in the cytoplasm to obtain the desired ratio.

### Focal adhesion coverage and filopodia and stress fiber quantification
HUVECs, subjected to siRNA treatment and transfected with GFP and lifeact-RFP constructs, were cultured to form a

confluent monolayer over 3 days. Subsequently, the cells were fixed and immunostained for paxillin and DAPI. High-resolution images of these monolayers were captured using a spinning disk confocal microscope.

To quantify stress fibers and filopodia, images were analyzed in Fiji (Schindelin et al., 2012), focusing specifically on cells expressing both GFP and lifeact-RFP. Stress fiber and filopodia were manually counted from maximal intensity projections of GFP and lifeact-RFP positive cells.

Focal adhesion coverage analysis was performed on carefully chosen z-planes that prominently displayed the focal adhesions. Initially, paxillin-stained images were processed with Fiji's "Remove Background" feature, applying a 50 px kernel. These images underwent segmentation via a custom-trained random forest classifier using LabKit (Arzt et al., 2022). Subsequently, cells positive for both GFP and lifeact-RFP were manually segmented. The coverage of focal adhesions was quantified from the LabKit segmentation output using Fiji's "Analyze Particles" function.

### Mitochondrial trapping experiments
Cells expressing the constructs of interest were plated on fibronectin-coated glass-bottom dishes (MatTek Corporation) for 2 h. Samples were fixed for 10 min using a solution of 4% PFA, then permeabilized using a solution of 0.25% (vol/vol) Triton X-100 for 3 min. Cells were washed with PBS and quenched using a solution of 1 M glycine for 30 min. Samples were washed three times in PBS and stored in PBS containing SiR-actin (100 nM; CY-SC001; Cytoskeleton) at 4°C until imaging. Just before imaging, samples were washed three times in PBS. Images were acquired using a spinning-disk confocal microscope (100× objective). The Pearson correlation coefficients were calculated from the 3D stacks using the JaCoP Fiji plugin (Schindelin et al., 2012; Bolte and Cordelières, 2006).

### Protein expression and purification
BL21(DE3)* competent cells were transformed with the relevant plasmid and grown in lysogeny broth supplemented with appropriate antibiotic (TLNRD1: 100 µg/ml ampicillin; CCM2: 50 µg/ml kanamycin) until the $OD_{600}$ reached ~0.6. Protein expression was induced by adding 0.4 mM IPTG and expressed overnight at 20°C. Cells were harvested by centrifugation, resuspended in lysis buffer (50 mM Tris-HCl pH 8, 250 mM NaCl, 5% vol/vol glycerol) at 5 ml/g of cells, and stored at –80°C.

TLNRD1 and SUMO-CCM2(413-438) proteins were purified using nickel-affinity chromatography and ion-exchange chromatography. Briefly, cells were thawed, supplemented with 1 mM PMSF, 5 mM DTT, 0.2% v/v Triton X-100, and lysed by sonication. Cell debris was removed by centrifugation, and the supernatant was filtered and loaded onto a 5 ml HisTrap HP column (Cytiva) using an AKTA Start (GE Healthcare). The column was washed in buffer containing 50 mM Tris-HCl pH 8, 600 mM NaCl, 30 mM imidazole, 4 mM $MgCl_2$, 4 mM ATP, 5% vol/vol glycerol, 0.2% vol/vol Triton X-100, and 5 mM DTT. Bound protein was eluted using a 75 ml linear gradient of 0–300 mM imidazole. Fractions containing the protein of interest were pooled and diluted with five volumes of either

20 mM sodium phosphate pH 6.5 (TLNRD1) or 20 mM Tri-HCl pH 8 (SUMO-CCM2[413-438]) and loaded onto either a 5 ml HiTrap SP (TLNRD1) or a 5 ml HiTrap Q (SUMO-CCM2) (Cytiva) and eluted with a 75-ml linear gradient of 0–750 mM NaCl. Purified proteins were dialyzed overnight against PBS pH 7.4 supplemented with 5 mM DTT, snap-frozen in $LN_2$, and stored at –80°C.

### GST pulldown assay
Cell pellets of the various GST-CCM2 constructs were thawed and lysed as above. The filtered supernatant was batch-bound to Glutathione Superflow Agarose resin (Thermo Fisher Scientific) for 90 min whilst being rolled at 4°C. Unbound proteins were removed by centrifugation/aspiration, and the resin was washed five times with 10 volumes of PBS pH 7.4. 50 µl washed resin containing immobilized GST-CCM2 was incubated with 200 µl purified protein at 40 µM for 60 min at room temperature with inversion mixing. The unbound protein was removed by centrifugation/aspiration. The resin was subjected to 5 × 500 µl washes of PBS pH 7.4, resuspended in 50 µl PBS pH 7.4, and analyzed by SDS-PAGE.

### Molecular modeling
To produce the structural models of the TLNRD1–CCM2 complex, the sequence of TLNRD1(FL) and CCM2(410-444) were submitted to the protein structure modeling tool, ColabFold (Mirdita et al., 2022), and the crystal structure of TLNRD1 (PDB accession no. 6XZ4) was used as a template. All figures were made using PyMOL (Version 2.5; Schrödinger, LLC).

### Fluorescence polarization assay
The SUMO-CCM2(413-438) contains a single cysteine (C437) that was used to couple the fusion protein with a maleimide-fluorescein dye (Thermo Fisher Scientific) following the manufacturer's protocol. Assays were performed in triplicate with a twofold serial protein dilution, with target peptides at 500 nM. FP was measured using a CLARIOstar plate reader (BMGLab-Tech) at 25°C (excitation: 482 ± 8 nm; emission: 530 ± 20 nm). Data were analyzed using GraphPad Prism 8 software, and $K_d$ values were generated using the one-site total binding equation.

### Actin preparation
Rabbit skeletal muscle acetone powder was kindly gifted by Professor Mike Geeves (University of Kent, Canterbury, UK), and actin was prepared using cycles of polymerization/depolymerization following the protocol of Spudich and Watt (Spudich and Watt, 1971). Briefly, 1 g of acetone powder was stirred on ice in extraction buffer (10 mM Tris pH 8, 0.5 mM ATP, 0.2 mM $CaCl_2$, 1 mM DTT), filtered, and clarified by ultracentrifugation (30,000 rpm, 70Ti rotor, 1 h). The supernatant was supplemented with 3 M KCl and 1 M $MgCl_2$ to final concentrations of 100 and 2 mM, respectively, stirred at room temperature for 1 h, and centrifuged for 3 h at 30,000 rpm. The pellet of F-actin was resuspended in a depolymerization buffer (5 mM Tris pH 7.5, 0.2 mM $CaCl_2$, 1 mM $NaN_3$), gently homogenized, and dialyzed against depolymerization buffer overnight. The dialyzed G-actin was centrifuged (30,000 rpm, 1 h), and the supernatant was

retained, supplemented with 0.2 mM ATP and 3% wt/vol sucrose, and snap-frozen in $LN_2$.

### F-actin cosedimentation assays

G-actin was polymerized by adding ATP, KCl, and $MgCl_2$ to final concentrations of 5 µM, 100 mM, and 2 mM, respectively. The polymerized actin was recovered by centrifugation (30,000 rpm, 1 h), gently homogenized in cosedimentation buffer (10 mM Tris pH 7.5, 100 mM NaCl, 2 mM $MgCl_2$, 1 mM DTT, 1 mM $NaN_3$), and stored at 4°C. Cosedimentation assays were performed with F-actin at 15 µM and equimolar concentrations of the other protein components. Binding was carried out at room temperature for 1 h, and the reaction was centrifuged at either 16,000 rpm (bundling assay) or 48,000 rpm (binding assay) in a TLA-100 ultracentrifuge rotor for 20 min. Equal volumes of supernatant and pellet fractions were analyzed by SDS-PAGE. Gels were quantified using ImageJ (Schindelin et al., 2012).

### Quantification and statistical analysis

Randomization tests were performed using the online tool PlotsOf-Differences (https://huygens.science.uva.nl/PlotsOfDifferences/) (Goedhart, 2019, *Preprint*). Dot plots were generated using PlotsOf-Data (Postma and Goedhart, 2019). Volcano plots were generated using VolcaNoseR (Goedhart and Luijsterburg, 2020). T-tests were performed using LibreOffice. When using t-tests, the distribution was assumed to be normal, but this was not formally tested.

### Online supplemental material

Fig. S1 shows the localization of TLNRD1, CCM2, PDCD10, and ITG1BP1 at the tip of MYO10 filopodia. Fig. S2 shows the efficiency of the tlnrd1 CRISPR guide and the expression of TLNRD1 in human embryos and in patients with CCM lesions. Fig. S3 shows additional data related to the role of TLNRD1 in endothelial cells. Fig. S4 shows additional data associated with the role of TLNRD1 in modulating endothelial barrier function. Fig. S5 shows additional data showing that TLNRD1 interacts with CCM2. Table S1 contains the mass spectrometry results of the TLNRD1 GFP-trap pulldowns. Table S2 contains the sequence of the plasmid and the primers generated in this study. Video 1 shows a ColabFold of the CCM2-TLNRD1 complex.

### Data availability

Plasmids generated in this study are being deposited in Addgene (https://www.addgene.org/Ben_Goult/ and https://www.addgene.org/Guillaume_Jacquemet/). The mass spectrometry proteomics data have been deposited to the ProteomeXchange Consortium via the PRIDE (Perez-Riverol et al., 2022) partner repository with the dataset identifier PXD045258. The raw microscopy images used to make the figures are available on Zenodo (https://doi.org/10.5281/zenodo.11185144, Ball et al., 2024). The authors declare that the data supporting this study's findings are available within the article and from the authors upon request. Any additional information required to reanalyze the data reported in this paper is available from the corresponding authors.

## Acknowledgments

The Cell Imaging and Cytometry Core facility, the Zebrafish Core (both at Turku Bioscience, University of Turku, Åbo Akademi University, and supported by Biocenter Finland), and Turku Bioimaging and the Genome Biology Unit (Research Programs Unit, HiLIFE Helsinki Institute of Life Science, Faculty of Medicine, University of Helsinki, Biocenter Finland) are acknowledged for services, instrumentation, and expertise. Mass spectrometry was performed at the Turku Proteomics Facility, University of Turku and Åbo Akademi University. The facility is supported by Biocenter Finland.

This study was supported by the Research Council of Finland (338537 to G. Jacquemet; 325464 to J. Ivaska; and 332402 to G. Follain), the Sigrid Juselius Foundation (to G. Jacquemet and to J. Ivaska), the Cancer Society of Finland (Syöpäjärjestöt; to G. Jacquemet and to J. Ivaska), and the Solutions for Health strategic funding to Åbo Akademi University (to G. Jacquemet). This research was supported by the InFLAMES Flagship Programme of the Academy of Finland (decision numbers: 337530, 337531, 357910, and 357911). This work was also supported by the Finnish Cancer Institute (K. Albin Johansson Professorship, J. Ivaska), the Centre of Excellence program (# 346131, J. Ivaska), and the Jane and Aatos Erkko Foundation (J. Ivaska). B.T. Goult was supported by the Biotechnology and Biological Sciences Research Council grant (BB/S007245/1) and the Cancer Research UK Program Grant (CRUK-A21671). The open access publication fees were funded by the Gösta Branders research fund, Åbo Akademi Research Foundation (Gösta Branders forskningsfond, Stiftelsen för Åbo Akademi).

Author contributions: N.J. Ball: Formal analysis, Investigation, Validation, Visualization, Writing—review & editing, S. Ghimire: Formal analysis, Investigation, Methodology, Writing—review & editing, G. Follain: Conceptualization, Data curation, Formal analysis, Investigation, Methodology, Visualization, Writing—original draft, Writing—review & editing, A. Pajari: Formal analysis, Investigation, Validation, D.H. Wurzinger: Investigation, Writing—review & editing, M. Vaitkeviciute: Investigation, A.R. Cowell: Conceptualization, Investigation, B. Berki: Investigation, J. Ivaska: Resources, Supervision, Writing—review & editing, I. Paatero: Formal analysis, Investigation, Methodology, Resources, Supervision, Visualization, Writing—review & editing, B.T. Goult: Conceptualization, Data curation, Formal analysis, Funding acquisition, Investigation, Methodology, Project administration, Resources, Supervision, Validation, Visualization, Writing—original draft, Writing—review & editing, G. Jacquemet: Conceptualization, Data curation, Formal analysis, Funding acquisition, Investigation, Methodology, Project administration, Resources, Software, Supervision, Validation, Visualization, Writing—original draft, Writing—review & editing.

Disclosures: All authors have completed and submitted the ICMJE Form for Disclosure of Potential Conflicts of Interest. I. Paatero reported "University of Turku has registered trademark 3DFLUOHISTO, and I. Paatero is involved in the commercialization of 3DFLUOHISTO-technology." No other disclosures were reported.

Submitted: 6 October 2023

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

# Supplemental material

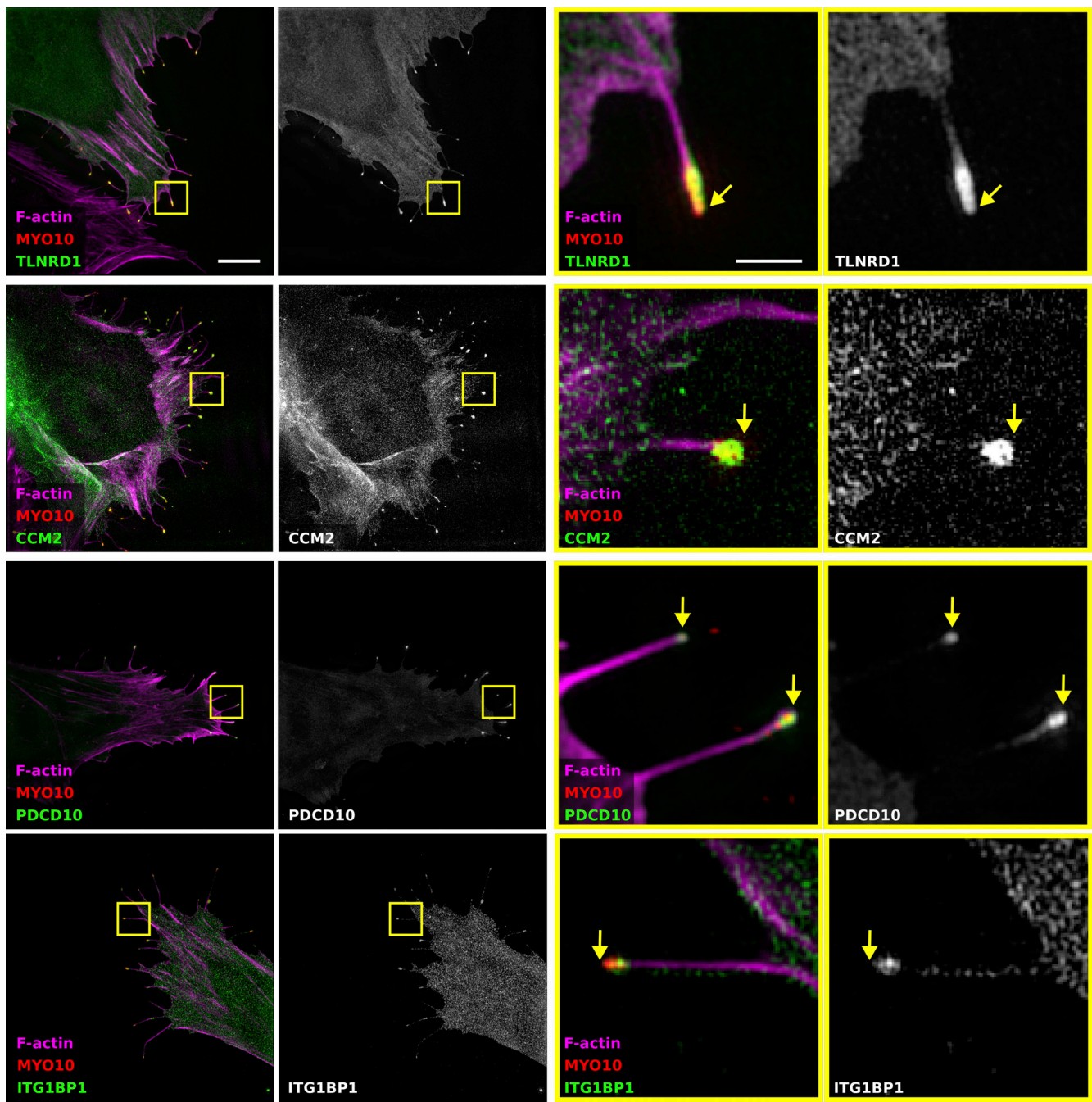

Figure S1.   **TLNRD1, CCM2, PDCD10, and ITG1BP1 localize at the tip of MYO10 filopodia.** U2OS cells expressing mScarlet-MYO10 with TLNRD1-GFP, CCM2-GFP, PDCD10-GFP, or ITG1BP1-GFP were plated on fibronectin for 2 h, fixed and stained to visualize F-actin. Samples were imaged using structured illumination microscopy. Representative maximum intensity projections are displayed; scale bars: (main) 5 µm; (inset) 1 µm. The yellow squares highlight magnified ROIs. The yellow arrows indicate the filopodia tips.

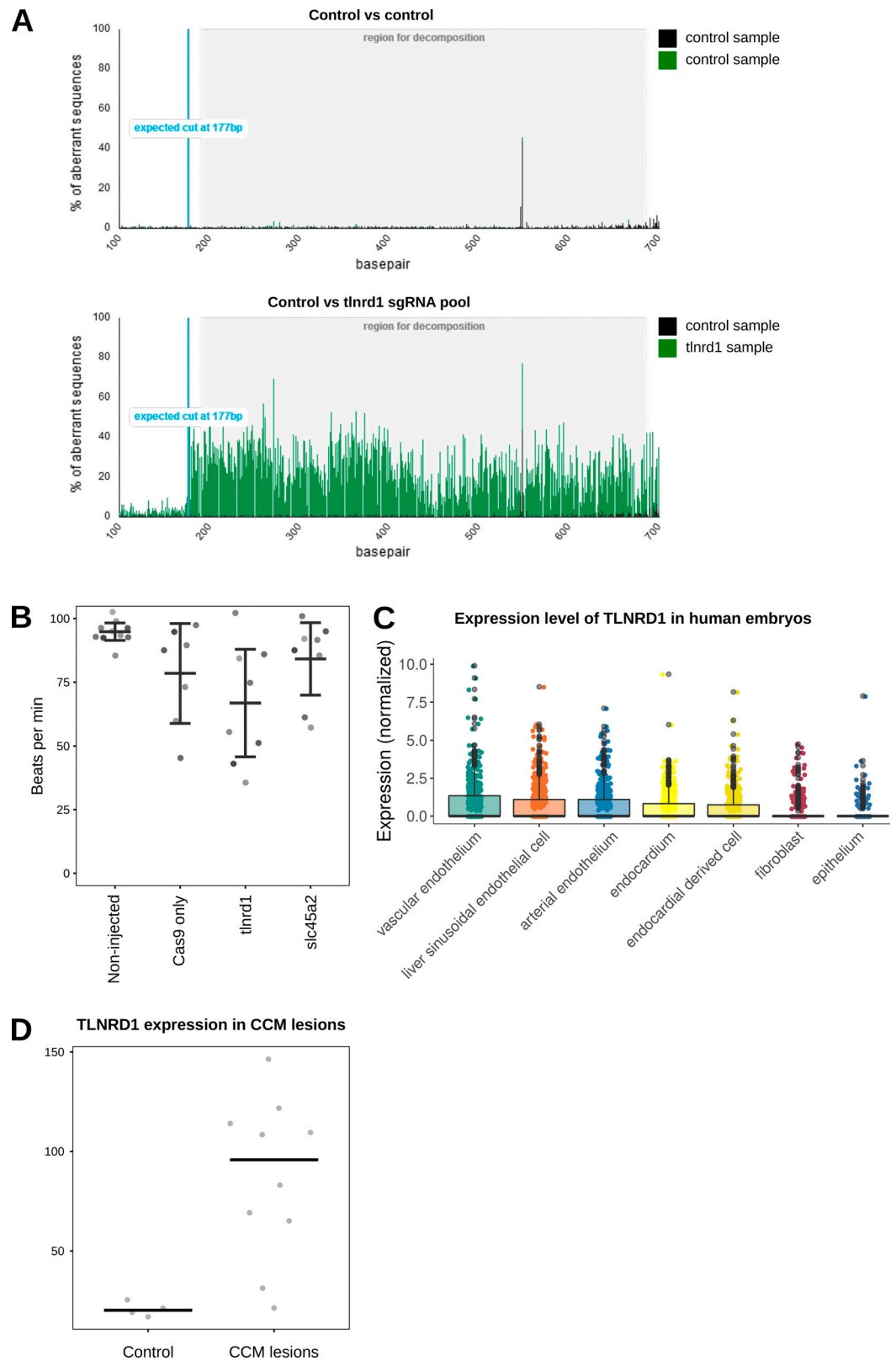

Figure S2. **TLNRD1 in vivo. (A)** Efficacy analysis of tlnrd1 CRISPR in zebrafish embryos. The Sanger sequencing chromatograms between control and tlnrd1 sgRNA injected samples were compared using TIDE software. The peak intensities show deviation of sequences and indicate effective editing of the tlnrd1 locus. **(B)** Zebrafish heart rate analysis. Zebrafish embryos were imaged using fast video microscopy, and heart rate was analyzed using kymographs in Fiji (*n* > 7 per condition). **(C)** TLNRD1 expression in human embryos in single cells in various endothelial compartments, fibroblasts, and epithelial cells (data from Xu et al., 2023). **(D)** TLNRD1 expression (RNA levels) in CCM lesions. Data from Subhash et al. (2019). Controls are four patients diagnosed with temporal lobe epilepsy.

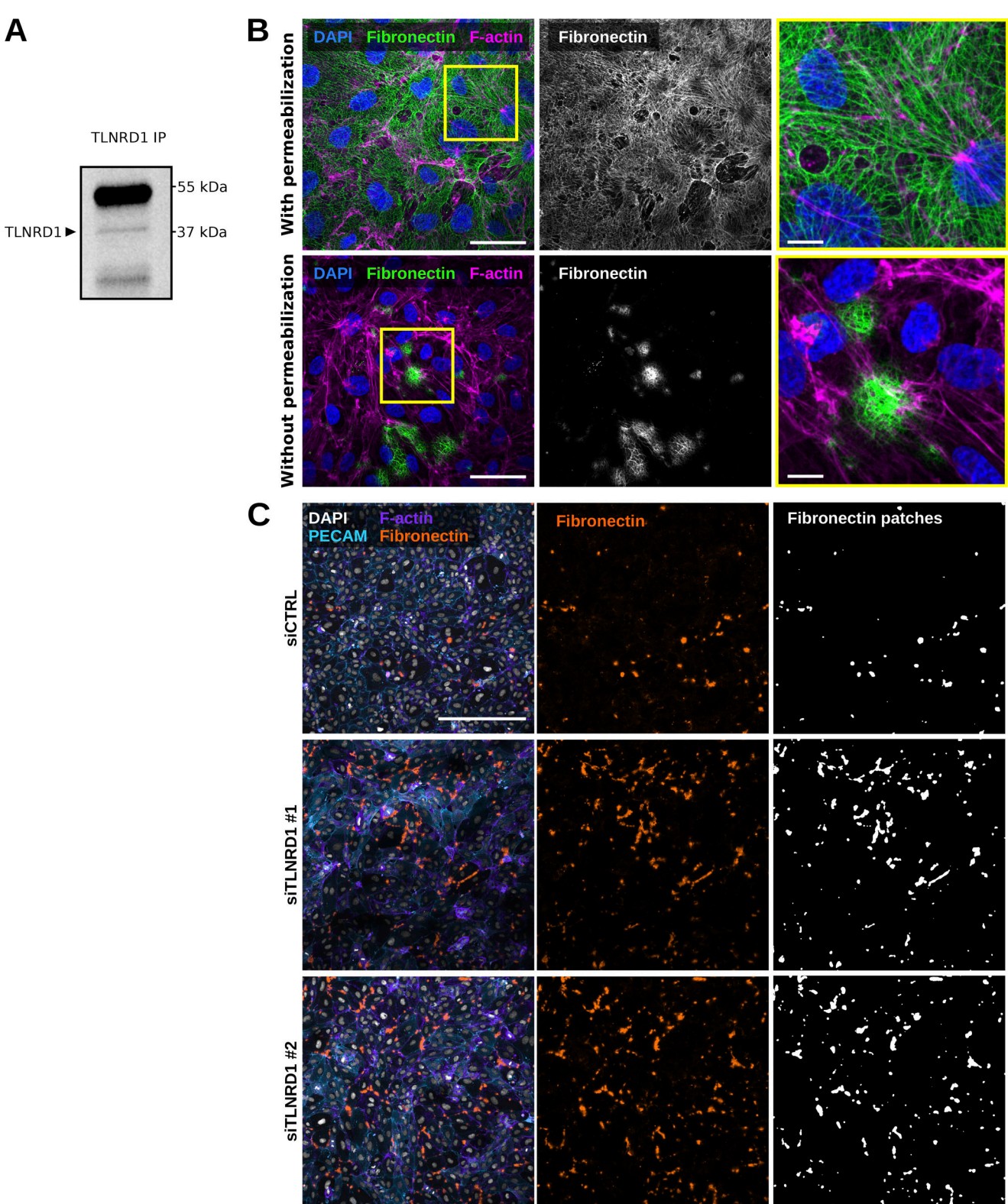

Figure S3. **TLNRD1 in endothelial cells. (A)** TLNRD1 immunoprecipitation from HUVEC lysate. A representative Western blot is displayed. **(B)** HUVECs were allowed to form a monolayer. Cells were then fixed and stained for DAPI, F-actin, and fibronectin (with or without permeabilization) before being imaged on a spinning disk confocal microscope. Representative maximum intensity projections are displayed. Scale bar: 250 µm. **(C)** TLNRD1 expression was silenced in HUVECs using two independent siRNA. HUVECs were then allowed to form a monolayer without flow stimulation. Cells were then fixed and stained for DAPI, F-actin, PECAM, and Fibronectin (without permeabilization) before being imaged on a spinning disk confocal microscope. Representative maximum intensity projections are displayed. Scale bar: 250 µm. Source data are available for this figure: SourceData FS3.

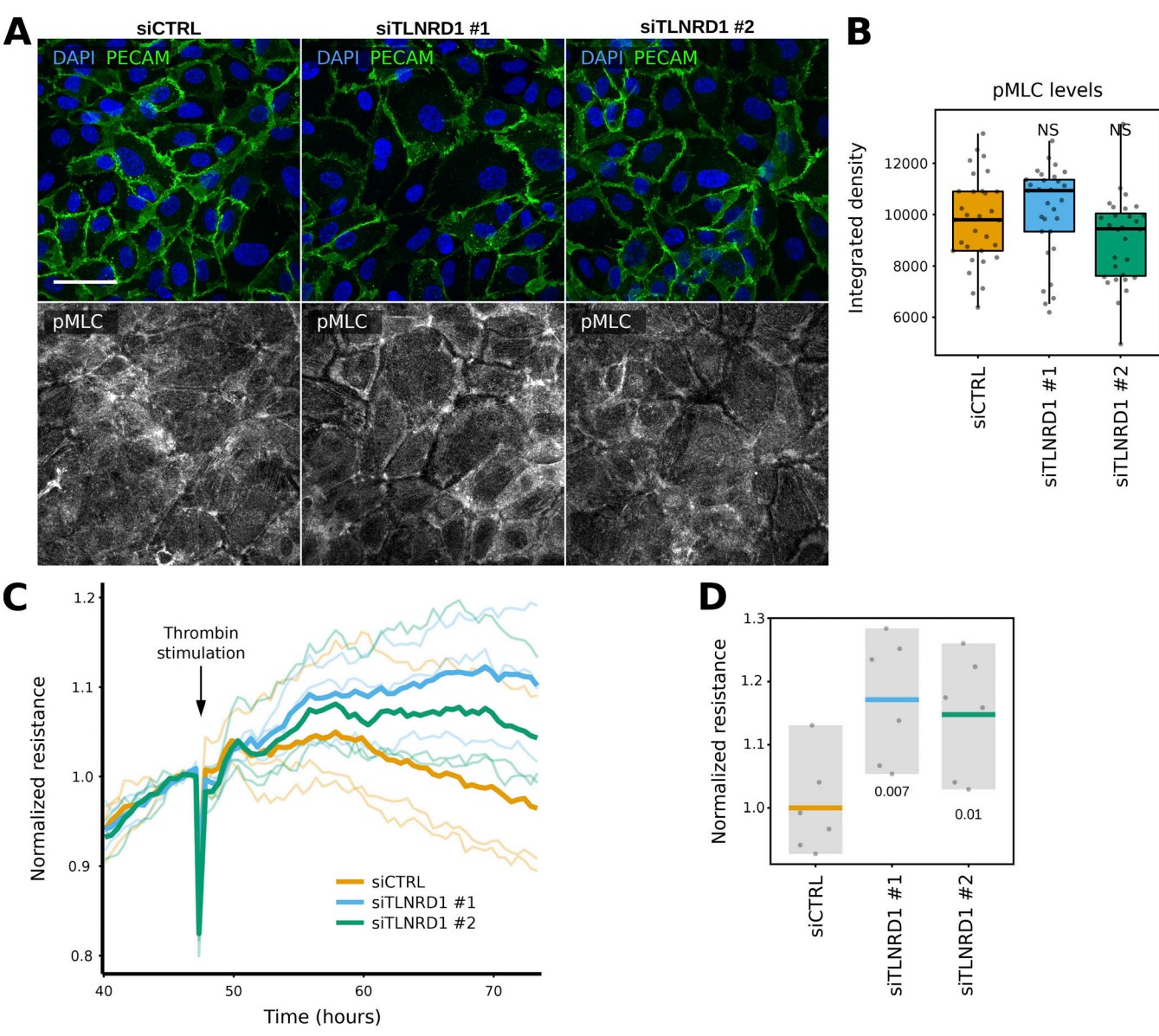

Figure S4. **TLNRD1 modulates endothelial barrier function. (A and B)** siCTRL and siTLNRD1 endothelial cells were allowed to form a monolayer without flow stimulation. Cells were then fixed and stained for phospho-Myosin light chain (pMLC S20) before being imaged on a spinning disk confocal microscope. **(A)** Representative sum projections are displayed. **(B)** The overall integrated density was measured for each field of view from SUM projections (three biological repeats, $n$ = 45 FOV per condition). **(C and D)** Assessment of trans-endothelial electrical resistance (TEER) in siCTRL and siTLNRD1 endothelial monolayers before and after thrombin stimulation was conducted utilizing the xCELLigence system. Individual TEER trajectories were normalized to the readings before the thrombin stimulation to study the effect of thrombin on TEER over time. Thrombin stimulation was performed 48 h after initial recording. **(C)** Displays representative data from one biological replicate. Here, the mean TEER trajectory from three individual wells is delineated with a bold line. In contrast, individual TEER curves are rendered in a lighter shade to delineate specific measurements within the same replicate. **(D)** Comparative analysis of the TEER values at 26 h after thrombin stimulation (two biological repeats, six measurements). The P values were determined using a $t$ test (two-sided, assuming unequal population variances).

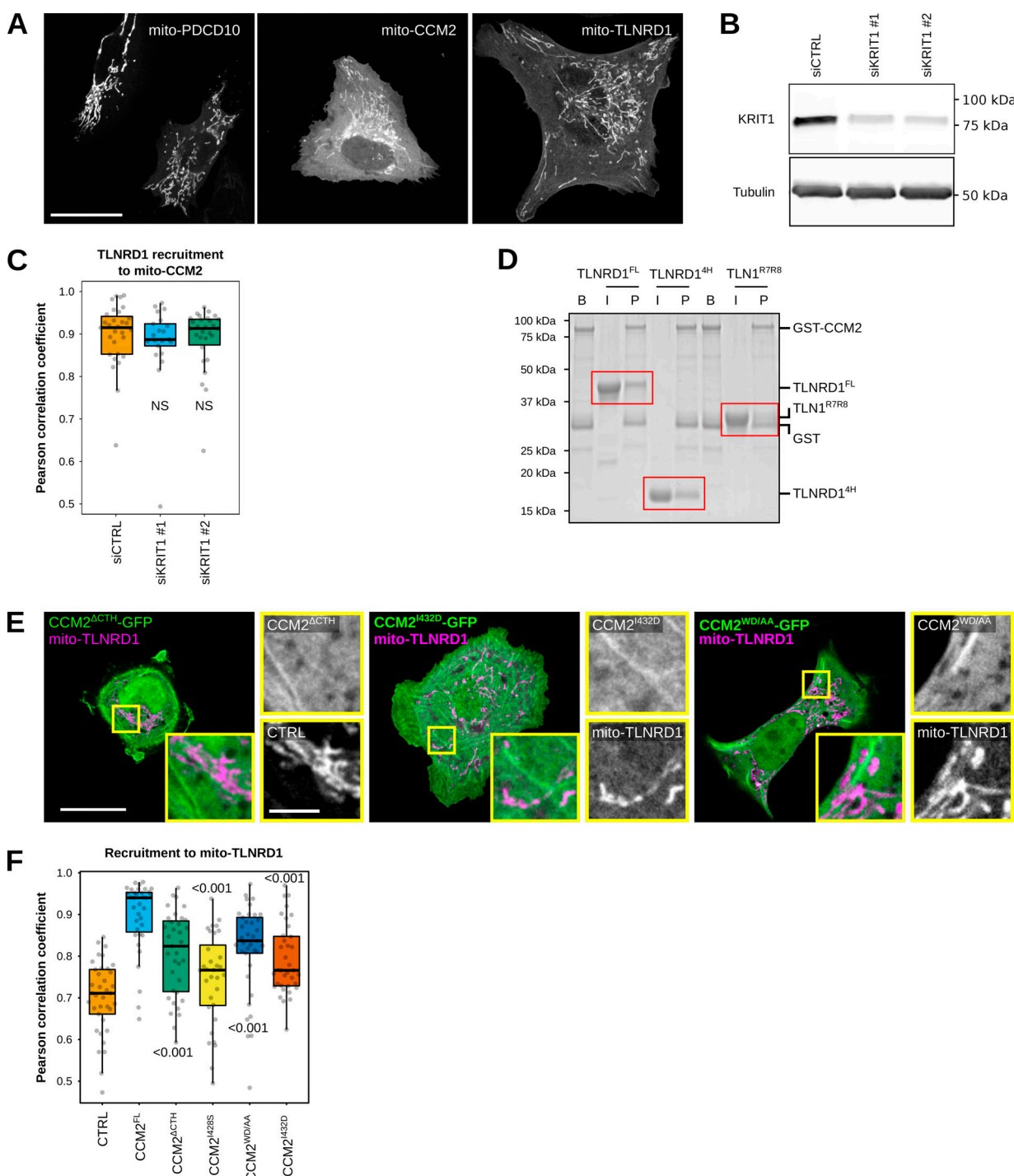

Figure S5. **TLNRD1 interacts with CCM2. (A)** U2OS cells expressing mito-PDCD10-mScarlet, mito-CCM2-mScarlet, or mito-TLNRD1-mScarlet were imaged using a spinning disk confocal microscope. **(B)** U2OS cells treated with siRNA targeting KRIT1 or siRNA control. KRIT1 levels were then analyzed using Western blots. A representative western blot is displayed. **(C)** U2OS cells treated with siRNA targeting KRIT1 or siRNA control and expressing TLNRD1-GFP and mito-CCM2-mScarlet were imaged using a spinning disk confocal microscope. 3D colocalization analyses were performed using the JACoP Fiji plugin, and results are displayed as Tukey boxplots (three biological repeats, $n > 21$ image stacks per condition). **(D)** Glutathione agarose-bound GST-CCM2 (beads: B) was incubated with recombinant TLNRD1, TLNRD1$^{4H}$, or TLN1$^{R7R8}$ (input: I). After multiple washes, proteins bound to the beads (pellet: P) were visualized. The various fractions were then analyzed using SDS-PAGE followed by Coomassie staining. A representative gel of three independent repeats is displayed. Red boxes highlight areas of interest in the gel. **(E)** U2OS cells expressing various GFP-tagged CCM2 constructs and mito-TLNRD1-mScarlet or mito-mScarlet (CTRL) were imaged using a spinning disk confocal microscope. Representative single Z-planes are displayed. The yellow squares highlight magnified ROIs. Scale bars: (main) 25 µm and (inset) 5 µm. **(F)** 3D colocalization analyses were performed using the JACoP Fiji plugin, and results are displayed as Tukey boxplots (three biological repeats, $n > 38$ image stacks per condition). The whiskers (shown here as vertical lines) extend to data points no further from the box than 1.5× the interquartile range. For all panels, the P values were determined using a randomization test. NS indicates no statistical difference between the mean values of the highlighted condition and the control. Source data are available for this figure: SourceData FS5.

Video 1: **ColabFold model of the TLNRD1–CCM2 complex.** Modeling of the TLNRD1–CCM2 complex using ColabFold using the TLNRD1 crystal structure (PDB accession no. 6XZ4) as a template. The TLNRD1 dimer is shown in blue and two CCM2 proteins are shown in yellow.

**Provided online are Table S1 and Table S2. Table S1 contains the mass spectrometry results of the TLNRD1 GFP-trap pulldowns. Table S2 contains the sequence of the plasmid and the primers generated in this study.**

