## [Peer Review File · The Journal of Cell Biology]

TLNRD1 is a CCM complex component and regulates endothelial barrier integrity

Neil Ball, Sujan Ghimire, Gautier Follain, Ada Pajari, Diana Wurzinger, Monika Vaitkevičiūtė, Alana Cowell, Bence Berki, Johanna Ivaska, Ilkka Paatero, Benjamin Goult, and Guillaume Jacquemet

Corresponding Author(s): Guillaume Jacquemet, Å...bo Akademi University

Review Timeline:

Submission Date:	2023-10-06
Editorial Decision:	2023-11-21
Revision Received:	2024-04-08
Editorial Decision:	2024-05-07
Revision Received:	2024-05-13

Monitoring Editor: Tatiana Petrova

Scientific Editor: Tim Fessenden

Transaction Report:

DOI: <https://doi.org/10.1083/jcb.202310030>

November 21, 2023

Re: JCB manuscript #202310030

Dr. Guillaume Jacquemet
Åbo Akademi University
Tykistökatu 6, 20520 Turku
Turku 20520
Finland

Dear Dr. Jacquemet,

Thank you for submitting your manuscript entitled "TLNRD1 is a CCM complex component and regulates endothelial barrier integrity". The manuscript has been evaluated by expert reviewers, whose reports are appended below. Unfortunately, after an assessment of the reviewer feedback, our editorial decision is against publication in JCB at this time.

You will see that, although both reviewers felt the topic and the results provided were intriguing, both also felt that major claims made were not sufficiently supported by the data shown and noted confirmation and controls needed to validate the findings. A suitably revised manuscript must include all changes requested by the reviewers, including some kind of confirmation of Rho/ROCK as noted by Reviewer 2.

Although your manuscript is intriguing, I feel that the points raised by the reviewers are more substantial than can be addressed in a typical revision period. If you wish to expedite publication of the current data, it may be best to pursue publication at another journal.

Given interest in the topic, I would be open to resubmission to JCB of a significantly revised and extended manuscript that fully addresses the reviewers' concerns and is subject to further peer-review. If you would like to resubmit this work to JCB, please contact the journal office to discuss an appeal of this decision or you may submit an appeal directly through our manuscript submission system. Please note that priority and novelty would be reassessed at resubmission.

Regardless of how you choose to proceed, we hope that the comments below will prove constructive as your work progresses. We would be happy to discuss the reviewer comments further once you've had a chance to consider the points raised in this letter. You can contact the journal office with any questions at cellbio@rockefeller.edu.

Thank you for thinking of JCB as an appropriate place to publish your work.

Sincerely,

Tatiana Petrova
Monitoring Editor
Journal of Cell Biology

Tim Fessenden
Scientific Editor
Journal of Cell Biology

Reviewer #1 (Comments to the Authors (Required)):

This manuscript by Ball et al. puts forward the concept that TLNRD1 is a novel identified component of the CCM complex and regulates endothelial barrier integrity. The biochemical data is convincing, and shows that TLNRD1 binds specifically to CCM2 and the authors pinpoint elegantly through which protein domains the interaction is mediated. The claims that TLNRD1 is a mediator of the CCM complex in the regulation of the actin cytoskeleton and the notion that TLNRD1 regulates the endothelial barrier are not convincingly shown, which limits my enthusiasms for the current manuscript at this point. Some of the results are exciting, and hold potential impact, however the data is presented somewhat preliminary and I would encourage to authors to further strengthen it.

Major feedback:

- Endothelial barrier function: the authors conclude that TLNRD1 modulates junctional integrity in endothelial cells. They conclude this on staining of fibronectin in non permeabilized endothelial layers. This is not necessarily a strong readout of endothelial barrier function. For instance, fibronectin is also deposited on the apical side of cells. To assess whether TLNRD1 controls barrier integrity it would be recommended to use direct permeability measurements (e.g. using fluorescent dextran; or measure resistance) and/or assess the junctional immunofluorescent stainings in more detail.
- The protein interaction studies show that there are specific residues in CCM2 and TLNRD1 that are needed for their interaction. The functional consequence of these mutants for the composition and function of the CCM complex in endothelial barrier function have not been investigated.
- The authors have knocked out TLNRD1 from zebrafish embryos using gRNA injections. Their major conclusion is that the vasculature is perturbed upon depletion. However, the data (Figure 2D) evidently shows that embryogenesis has been defective, suggesting an important role for TLNRD1 in other tissues. A more unbiased assessment of the phenotype of these fish would be needed to understand the importance of TLNRD1 in zebrafish development. Also, it is not entirely clear why slc45a2 injected embryos were shown as control condition. Please explain.

Minor feedback:

- Some of the interaction data would be stronger if also quantified from the independent experiments (for example in Figures 4F, S4D).
- Some things were unclear from the mass spectrometry data: why do most of the significant TLNRD1-GFP interactin proteins have exactly the same fold change?
- The authors present data that shows that TLNRD1 expression is increased in CCM lesions from patients. It is unclear what and how the patient material was investigated. Was this based on RNA or protein?
- Single cell seq data in Figure 2AB: what is the evidence that the indicated populations are endothelial cells? Which markers were used to mark these cells?
- Can you explain in a little more detail how the fluorescent polarization assay works?
- Figure 4A: why was CCM1 localization not included in the interaction studies in Figure 4?

Reviewer #2 (Comments to the Authors (Required)):

The authors found that the recently identified protein TLNRD1 is primarily expressed in the vasculature in vivo. Its depletion leads to vascular abnormalities in vivo and loss of barrier integrity in cultured endothelial cells. Furthermore, The authors show that TLNRD1 is part of the CMM complex and that biochemical precipitation assays identify CMM2 as a binding partner and link this to vascular dementia.

Overall, this is a classic and thorough cell biology study. It shows how the protein interacts with a protein complex, and detects the localization in animal and cell models, using endogenous stainings and overexpression.

The quality of the pull-downs are of high quality and the modeling helps to understand where the proteins bind and how the complex may behave. The immunofluorescence of the HUVEC cultures can be improved. In particular, their claim that TLNRD1 resides at junctions is not supported. Please include a co-localization with VE-cadherin, as the majority of the protein seems to localize in the cytosol.

For the KD studies, they show reduced mRNA levels, but staining protein levels would be better, in particular, as the cells show a different morphology upon KD, one would like to be sure actin/tubulin are still okay, and the target protein expression is diminished.

Based on only a single f-actin-stained image, it is too much of a stretch to state that the target protein controls the actin cytoskeleton. Can the authors include focal adhesion stains and study in more detail cortical actin bundles upon KD? This would argue for a mechanistic role in the regulation of junctions, i.e. EC integrity. The overexpression studies are important and informative, as shown in Fig 6, but KD would show the actin cytoskeleton regulation even better.

For the permeability test, using the FN patches, it would be better to include a true permeability/leaky assay, for example, a Transwell assay or measuring the electrical resistance over the EC monolayer. Adding these data would increase the strength of the paper.

The final statement that TLNRD1 works independently from Rho/Rho kinase is not supported by any data, only by speculation. I would either add one or two experiments showing this (using the inhibitor C3 or Y27632) or tone down this remark.

Reviewer #1 (Comments to the Authors (Required)):

This manuscript by Ball et al. puts forward the concept that TLNRD1 is a novel identified component of the CCM complex and regulates endothelial barrier integrity. The biochemical data is convincing, and shows that TLNRD1 binds specifically to CCM2 and the authors pinpoint elegantly through which protein domains the interaction is mediated. The claims that TLNRD1 is a mediator of the CCM complex in the regulation of the actin cytoskeleton and the notion that TLNRD1 regulates the endothelial barrier are not convincingly shown, which limits my enthousiams for the current manuscript at this point. Some of the results are exciting, and hold potential impact, however the data is presented somewhat preliminary and I would encourage to authors to further strengthen it.

Thank you for acknowledging the strengths of our biochemical data, particularly in demonstrating the specific binding of TLNRD1 to CCM2 and identifying the protein domains involved in this interaction. We appreciate your recognition of the novelty and potential impact of our findings. We understand your concern regarding the evidence supporting TLNRD1's role as a mediator in regulating the actin cytoskeleton and its influence on endothelial barrier integrity in the original manuscript. Following your feedback, we have taken significant steps to strengthen this aspect of our study (see answers below). We hope that our revisions meet your expectations.

Major feedback:

- Endothelial barrier function: the authors conclude that TLNRD1 modulates junctional integrity in endothelial cells. They conclude this on staining of fibronectin in non permeabilized endothelial layers. This is not necessarily a strong readout of endothelial barrier function. For instance, fibronectin is also deposited on the apical side of cells. To assess whether TLNRD1 controls barrier integrity it would be recommended to use direct permeability measurements (e.g. using fluorescent dextran; or measure resistance) and/or assess the junctional immunofluorescent stainings in more detail.

Your point about the limitations of solely relying on fibronectin staining in non-permeabilized endothelial layers is well-taken. It is, however, important to specify that we never observed fibronectin behind deposited apically in our experiments. Nevertheless, we recognize the need for more direct and robust assays to assess endothelial barrier function and have now incorporated additional experiments into our revised manuscript. In particular, we performed Trans-endothelial electrical resistance (TEER) measurements:

To assess the integrity of the endothelial barrier directly, we first conducted TEER measurements during the formation of endothelial monolayers. These experiments revealed that TLNRD1 silencing significantly delays the achievement of maximal TEER values, indicating defects in the assembly of an impermeable monolayer. This result corroborates our hypothesis that TLNRD1 plays a crucial role in endothelial barrier formation (Refer to Fig. 3I and 3J in our revised manuscript and below for your convenience).

Figure 3. (I-J) Assessment of trans-endothelial electrical resistance (TEER) in siCTRL and siTLNRD1 endothelial monolayers was conducted utilizing the xCELLigence system. Individual TEER trajectories were normalized to their final readings to study the establishment of the TEER over time. (I) Displays representative data from one biological replicate. Here, the mean TEER trajectory from three individual wells is delineated with a bold line. In contrast, individual TEER curves are rendered in a lighter shade to delineate specific measurements within the same replicate. (J) Focuses on the comparative analysis at the time when siCTRL cells attain 70% of their ultimate TEER values, highlighting the impact of TLNRD1 silencing on developing endothelial barrier function (4 biological repeats, 11 measurements).

Next, to explore TLNRD1's role beyond the initial formation of the endothelial barrier, we investigated its contribution to barrier function modulation in established monolayers subjected to inflammatory conditions. Specifically, we treated these monolayers with thrombin, a protease known to induce endothelial barrier dysfunction. Intriguingly, TLNRD1 silencing resulted in an enhanced barrier function following thrombin treatment. This finding aligns with recent work (Schnitzler et al., 2024) and underscores TLNRD1's critical involvement in modulating endothelial permeability under inflammatory stimuli (Refer to Supplementary Figures S4C and S4D and below for your convenience).

Altogether, our data indicate that TLNRD1 contributes to establishing and regulating endothelial monolayer permeability; it initially contributes to forming an impermeable monolayer, but subsequently, TLNRD1 modulates the permeability in the established monolayer.

Figure S4. (C-D) Assessment of trans-endothelial electrical resistance (TEER) in siCTRL and siTLNRD1 endothelial monolayers before and after thrombin stimulation was conducted utilizing the xCELLigence system. Individual TEER trajectories were normalized to the readings before the thrombin stimulation to study the effect of thrombin on TEER over time. Thrombin stimulation was performed 48 hours post-initial recording. **(C)** Displays representative data from one biological replicate. Here, the mean TEER trajectory from three individual wells is delineated with a bold line. In contrast, individual TEER curves are rendered in a lighter shade to delineate specific measurements within the same replicate. **(D)** Comparative analysis of the TEER values at 26h post-thrombin stimulation (2 biological repeats, 6 measurements).

- The protein interaction studies show that there are specific residues in CCM2 and TLNRD1 that are needed for their interaction. The functional consequence of these mutants for the composition and function of the CCM complex in endothelial barrier function have not been investigated.

Thank you for highlighting the need for a deeper investigation into the functional consequences of the TLNRD1-CCM2 interaction, particularly about the function of the CCM complex in endothelial barrier function. We acknowledge the challenge this presents, especially when working with primary cells. To date, the role of the CCM complex has never been investigated (only the individual components). As we have outlined in our manuscript, the CCM2 mutants identified exhibit additional, complex phenotypes that extend beyond the scope of TLNRD1's involvement. This complexity limits our ability to use these mutants directly to dissect the specific role of the TLNRD1-CCM2 interaction in endothelial barrier function directly.

In response to your and reviewer 2 feedback, we designed experiments to explore how disruptions in the TLNRD1-CCM2 interaction influence endothelial cell behavior, focusing on actin cytoskeleton organization and focal adhesion formation. We focused on actin stress fiber formation (a phenotype associated with the CCM complex and known to contribute to endothelial barrier function) and filopodia formation (a phenotype associated with known TLNRD1 function). We employed a targeted approach by silencing TLNRD1 and then expressing various constructs to dissect the functional implications of this interaction. The results are presented in Figure 7 of our updated manuscript.

- Wild-type TLNRD1 (TLNRD1) for assessing the restoration of function upon TLNRD1 reintroduction.
- TLNRD1^{2T} Mutant to evaluate the effects of disrupting the TLNRD1-CCM2 interaction.
- TLNRD1^{F250D} Mutant to explore the impact of hindering TLNRD1 dimerization, previously shown to abolish its actin-bundling activity.

These experiments revealed that TLNRD1 silencing altered endothelial cell morphology, evidenced by expanded cell areas, augmented stress fiber counts, increased paxillin-positive adhesion coverage, and diminished filopodia formation. This underscored TLNRD1's pivotal role in cytoskeletal organization and cell adhesion. Reintroducing TLNRD1 into silenced cells reverted these alterations, confirming TLNRD1's essential role in preserving endothelial cell architecture and dynamics. Meanwhile, the TLNRD1^{F250D} expression failed to reverse these effects, highlighting the critical need for a complete dimerization motif in TLNRD1 for its full functionality. Interestingly, expression of the TLNRD1^{2T} mutant partially restored the TLNRD1 silencing phenotype, specifically fully restoring filopodia formation but not completely reversing other cytoskeletal and adhesion-related changes.

Our findings highlight the critical role of the TLNRD1-CCM2 interaction in the modulation of actin stress fiber and focal adhesion formation in endothelial cells. The necessity of an intact dimerization motif in TLNRD1 for these phenotypes, coupled with the observation that TLNRD1 interactions with actin and CCM2 are mutually exclusive, suggests that TLNRD1's cellular roles are likely context-dependent. We speculate that in environments rich in F-actin, TLNRD1 preferentially binds and bundles F-actin, facilitating processes such as filopodia formation. Conversely, in the presence of CCM2, TLNRD1 pivots towards supporting the CCM complex's function, potentially aiding in the formation of CCM complex multimers and the assembly of signaling platforms. Our data also do not exclude the possibility that TLNRD1 may act as a molecular bridge, linking the CCM complex with the actin cytoskeleton. Future investigations will be aimed at further deciphering TLNRD1's contributions to CCM complex assembly.

Figure 7: CCM2 modulates TLNRD1 localization and bundling activity

(A-B) HUVECs treated with siCTRL or siTLNRD1 siRNA and expressing lifact-RFP along with GFP, TLNRD1-GFP, TLNRD1^{2T}-GFP, or TLNRD1^{F250D}-GFP, were fixed and stained for DAPI and paxillin, followed by imaging using a

spinning disk confocal microscope. (A) Maximal intensity projections of representative fields of view. Highlighted within yellow squares are ROIs selected for magnification. The upper ROI panel presents maximal intensity projections showcasing Lifeact-RFP and GFP-positive cells. In contrast, the lower ROI panels concentrate on the basal plane to showcase the paxillin-positive adhesions, with yellow outlines delineating the contours of GFP-positive cells. Scale bars: (main) 50 μm and (inset) 20 μm . (B) Quantitative analysis was performed on GFP-positive cells, evaluating various parameters: cell area, the proportion of the cell area covered by paxillin-positive adhesions, and the number of actin stress fibers and of filopodia (4 biological repeats, >60 fields of view per condition). The results are shown as Tukey boxplots. The whiskers (shown here as vertical lines) extend to data points no further from the box than 1.5 \times the interquartile range. The p-values were determined using a randomization test.

- The authors have knocked out TLNRD1 from zebrafish embryos using gRNA injections. Their major conclusion is that the vasculature is perturbed upon depletion. However, the data (Figure 2D) evidently shows that embryogenesis has been defective, suggesting an important role for TLNRD1 in other tissues. A more unbiased assessment of the phenotype of these fish would be needed to understand the importance of TLNRD1 in zebrafish development. Also, it is not entirely clear why slc45a2 injected embryos were shown as control condition. Please explain.

We recognize the necessity of a more nuanced interpretation of our findings. The perturbations observed in embryogenesis could indeed suggest a possible, significant role for TLNRD1 in other tissues in addition to its impact on vasculature development.

In particular, we write, "While the observed effects of TLNRD1 depletion in zebrafish embryos could be attributed to TLNRD1 functions beyond the vascular system, our findings suggest a significant role for TLNRD1 in vascular regulation in vivo."

The revised manuscript provides a more detailed explanation of using slc45a2-injected embryos as an additional control condition. In the method section, we write:

"The slc45a2 gene encodes a transporter protein that is crucial in melanin synthesis. In zebrafish, the disruption of the slc45a2 function leads to a visible loss of pigmentation in the eyes, an easily observable and quantifiable phenotype. We utilized slc45a2 as an additional control in our experiments because it is a reliable positive control for CRISPR-mediated effects that is often used (Heliste et al., 2020). The loss of pigmentation in embryos where slc45a2 is disrupted is a clear indicator of successful genome editing".

Minor feedback:

- Some of the interaction data would be stronger if also quantified from the independent experiments (for example in Figures 4F, S4D).

We thank the reviewer for raising this point. The experiments presented in the figures mentioned by the reviewer are qualitative in nature, and we now emphasize this in the text. We write:

"These assays were only qualitative, so to quantify this interaction, we used a Fluorescence Polarization (FP) assay, in which unlabelled TLNRD1 was titrated against a fixed concentration

of fluorescein-labeled CCM2(413-438) peptide. The FP assay revealed that TLNRD1 and TLNRD14H interact with CCM2⁴¹³⁻⁴³⁸ with a dissociation constant (Kd) in the nanomolar range”

- Some things were unclear from the mass spectrometry data: why do most of the significant TLNRD1-GFP interactin proteins have exactly the same fold change?

The uniformity in fold change observed for the significant interacting proteins can be attributed to their unique presence in the TLNRD1-GFP condition compared to the control. In our data analysis process, a large-fold change value is assigned uniformly to these unique interactions to visualize these proteins on the volcano plot.

We now indicate in the figure legend:

“Notably, proteins uniquely identified in either the TLNRD1 or GFP conditions were assigned a fold change of 400 to be displayed on the volcano plot.”

- The authors present data that shows that TLNRD1 expression is increased in CCM lesions from patients. It is unclear what and how the patient material was investigated. Was this based on RNA or protein?

In this case, it is based on RNA. We now write in the manuscript:

“Importantly, a reanalysis of publicly available datasets revealed that TLNRD1 expression at the RNA level is up-regulated in CCM lesions in patients”

- Single cell seq data in Figure 2AB: what is the evidence that the indicated populations are endothelial cells? Which markers were used to mark these cells?

These analyses were performed by (Schaum et al., 2018) in the Tabula Muris project. They report that for the brain, endothelial cells were defined as Cdh5+, Pecam1+, Slco1c1+, and Ocln+; in the heart, endothelial cells were defined as Cdh5+ and Pecam1+.

We now provide this information in the figure legend.

- Can you explain in a little more detail how the fluorescent polarization assay works?

In the updated text, we now write:

“These assays were only qualitative, so to quantify this interaction, we used a Fluorescence Polarization (FP) assay, in which unlabelled TLNRD1 was titrated against a fixed concentration of fluorescein-labeled CCM2(413-438) peptide. The FP assay revealed that TLNRD1 and

TLNRD14H interact with CCM2413-438 with a dissociation constant (Kd) in the nanomolar range.”

- Figure 4A: why was CCM1 localization not included in the interaction studies in Figure 4?

We originally included CCM1 in the experimental plan. However, when we generated the CCM1 construct to target CCM1 to the mitochondria, it did not localize properly. Therefore, to assess CCM1's contribution, we instead used siRNA to test if silencing CCM1 would disrupt the recruitment of TLNRD1 to CCM2 (Fig. S5).

Reviewer #2 (Comments to the Authors (Required)):

The authors found that the recently identified protein TLNRD1 is primarily expressed in the vasculature in vivo. Its depletion leads to vascular abnormalities in vivo and loss of barrier integrity in cultured endothelial cells. Furthermore, The authors show that TLNRD1 is part of the CMM complex and that biochemical precipitation assays identify CMM2 as a binding partner and link this to vascular dementia.

Overall, this is a classic and thorough cell biology study. It shows how the protein interacts with a protein complex, and detects the localization in animal and cell models, using endogenous stainings and overexpression. The quality of the pull-downs are of high quality and the modeling helps to understand where the proteins bind and how the complex may behave.

We are sincerely grateful for your encouraging and positive feedback on our study of the protein TLNRD1, its expression in the vasculature, and its role in vascular abnormalities and endothelial barrier integrity.

The immunofluorescence of the HUVEC cultures can be improved. In particular, their claim that TLNRD1 resides at junctions is not supported. Please include a co-localization with VE-cadherin, as the majority of the protein seems to localize in the cytosol.

We agree with the reviewer. TLNRD1 does not localize to cell-cell junctions, as we indicated in the original manuscript:

”In HUVEC monolayers, TLNRD1 localized to the cytoplasm and accumulated on actin bundles, including stress fibers and filopodia (Fig. 3A). TLNRD1 also localized on the actin structures near cell-cell junctions (Fig. 3A).”

To further clarify, we now include, “However, we did not observe an accumulation of TLNRD1 at cell-cell junctions.”

For the KD studies, they show reduced mRNA levels, but staining protein levels would be better, in particular, as the cells show a different morphology upon KD, one would like to be sure actin/tubulin are still okay, and the target protein expression is diminished.

We thank the reviewer for raising this point. As indicated in the manuscript, both western blots and staining of endogenous TLNRD1 levels in endothelial cells are highly challenging. This is why we use qPCR to assess the efficacy of our siRNA knockdown.

Importantly, we now include rescue experiments in the revised manuscript by re-expressing WT TLNRD1, validating our silencing strategy (see Figure 7).

Based on only a single f-actin-stained image, it is too much of a stretch to state that the target protein controls the actin cytoskeleton. Can the authors include focal adhesion stains and study in more detail cortical actin bundles upon KD? This would argue for a mechanistic role in the regulation of junctions, i.e. EC integrity. The overexpression studies are important and informative, as shown in Fig 6, but KD would show the actin cytoskeleton regulation even better.

We thank the reviewer for this great suggestion; the new data we have added to address this has significantly strengthened the paper.

To address this point and a point raised by reviewer 1, TLNRD1-silenced cells were transfected with various constructs, including; i) GFP as a control, ii) TLNRD1 wild type to evaluate functional restoration, iii) TLNRD1^{2T} to assess the impact of disrupting the TLNRD1-CCM2 interaction, and iv) TLNRD1^{F250D} to investigate the effects of blocking TLNRD1's dimerization capabilities. Cells were also co-transfected with lifeact-RFP to enable a detailed visualization of the actin cytoskeleton in individual cells within the monolayer. Paxillin staining was employed to visualize focal adhesions (Fig. 7A and 7B).

These experiments revealed that TLNRD1 silencing alters endothelial cell morphology, evidenced by expanded cell areas, augmented stress fiber counts, increased paxillin-positive adhesion coverage, and diminished filopodia formation. These findings underscore TLNRD1's pivotal role in cytoskeletal organization and cell adhesion. Reintroducing TLNRD1 into silenced cells reverted these alterations, confirming TLNRD1's essential role in preserving endothelial cell architecture and dynamics.

However, what was striking was that the reintroduction of the TLNRD1 constructs with our well-characterized mutations gave different outcomes. The TLNRD1^{F250D} expression failed to reverse these effects, highlighting the critical need for a complete dimerization motif in TLNRD1 for its full functionality. Interestingly, the TLNRD1^{2T} mutant's expression restored part of the TLNRD1 silencing phenotype; specifically, it fully restored filopodia formation but did not completely reverse the other cytoskeletal and adhesion-related changes.

Our findings highlight the critical role of the TLNRD1-CCM2 interaction in the modulation of actin stress fiber and focal adhesion formation in endothelial cells. The necessity of an intact dimerization motif in TLNRD1 for these phenotypes, coupled with the observation that TLNRD1

interactions with actin and CCM2 are mutually exclusive, suggests that TLNRD1's cellular roles are likely context-dependent. We speculate that in environments rich in F-actin, TLNRD1 preferentially binds and bundles F-actin, facilitating processes such as filopodia formation. Conversely, in the presence of CCM2, TLNRD1 pivots towards supporting the CCM complex's function, potentially aiding in the formation of CCM complex multimers and the assembly of signaling platforms. Our data also do not exclude the possibility that TLNRD1 may act as a molecular bridge, linking the CCM complex with the actin cytoskeleton. Future investigations will be aimed at further deciphering TLNRD1's contributions to CCM complex assembly.

Figure 7: CCM2 modulates TLNRD1 localization and bundling activity

(A-B) HUVECs treated with siCTRL or siTLNRD1 siRNA and expressing lifeact-RFP along with GFP, TLNRD1-GFP, TLNRD1^{2T}-GFP, or TLNRD1^{F250D}-GFP, were fixed and stained for DAPI and paxillin, followed by imaging using a spinning disk confocal microscope. (A) Maximal intensity projections of representative fields of view. Highlighted

within yellow squares are ROIs selected for magnification. The upper ROI panel presents maximal intensity projections showcasing lifeact-RFP and GFP-positive cells. In contrast, the lower ROI panels concentrate on the basal plane to showcase the paxillin-positive adhesions, with yellow outlines delineating the contours of GFP-positive cells. Scale bars: (main) 50 μm and (inset) 20 μm . (B) Quantitative analysis was performed on GFP-positive cells, evaluating various parameters: cell area, the proportion of the cell area covered by paxillin-positive adhesions, and the number of actin stress fibers and of filopodia (4 biological repeats, >60 fields of view per condition). The results are shown as Tukey boxplots. The whiskers (shown here as vertical lines) extend to data points no further from the box than 1.5 \times the interquartile range. The p-values were determined using a randomization test.

For the permeability test, using the FN patches, it would be better to include a true permeability/leaky assay, for example, a Transwell assay or measuring the electrical resistance over the EC monolayer. Adding these data would increase the strength of the paper.

As also mentioned to Reviewer 1, To assess the integrity of the endothelial barrier directly, we first conducted TEER measurements during the formation of endothelial monolayers. These experiments revealed that TLNRD1 silencing significantly delays the achievement of maximal TEER values, indicating defects in the assembly of an impermeable monolayer. This quantitative approach provides a more direct assessment of barrier integrity. It corroborates our hypothesis that TLNRD1 plays a crucial role in endothelial barrier formation (Refer to Fig. 3I and 3J in our revised manuscript and below for your convenience).

Figure 3. (I-J) Assessment of trans-endothelial electrical resistance (TEER) in siCTRL and siTLNRD1 endothelial monolayers was conducted utilizing the xCELLigence system. Individual TEER trajectories were normalized to their final readings to study the establishment of the TEER over time. (I) Displays representative data from one biological replicate. Here, the mean TEER trajectory from three individual wells is delineated with a bold line. In contrast, individual TEER curves are rendered in a lighter shade to delineate specific measurements within the same replicate. (J) Focuses on the comparative analysis at the time when siCTRL cells attain 70% of their ultimate TEER values, highlighting the impact of TLNRD1 silencing on developing endothelial barrier function (4 biological repeats, 11 measurements).

To explore TLNRD1's role beyond the initial formation of the endothelial barrier, we also investigated its contribution to barrier function modulation in established monolayers subjected to inflammatory conditions. Specifically, we treated these monolayers with thrombin, a protease

known to induce endothelial barrier dysfunction. Intriguingly, TLNRD1 silencing resulted in an enhanced barrier function following thrombin treatment. This finding aligns with recent work (Schnitzler et al., 2024) and underscores TLNRD1's critical involvement in modulating endothelial permeability under inflammatory stimuli (Refer to Supplementary Figures S4C and S4D and below for your convenience).

Altogether, our data indicate that TLNRD1 contributes to establishing and regulating endothelial monolayer permeability; it initially contributes to forming an impermeable monolayer, but subsequently, TLNRD1 modulates the permeability in the established monolayer.

Figure S4. (C-D) Assessment of trans-endothelial electrical resistance (TEER) in siCTRL and siTLNRD1 endothelial monolayers before and after thrombin stimulation was conducted utilizing the xCELLigence system. Individual TEER trajectories were normalized to the readings before the thrombin stimulation to study the effect of thrombin on TEER over time. Thrombin stimulation was performed 48 hours post-initial recording. (C) Displays representative data from one biological replicate. Here, the mean TEER trajectory from three individual wells is delineated with a bold line. In contrast, individual TEER curves are rendered in a lighter shade to delineate specific measurements within the same replicate. (D) Comparative analysis of the TEER values at 26h post-thrombin stimulation (2 biological repeats, 6 measurements).

The final statement that TLNRD1 works independently from Rho/Rho kinase is not supported by any data, only by speculation. I would either add one or two experiments showing this (using the inhibitor C3 or Y27632) or tone down this remark.

In our initial submission, we referenced supporting data indicating that phospho-Myosin Light Chain (pMLC) levels were not up-regulated upon TLNRD1 silencing, in contrast to what is typically observed with CCM2 silencing. These findings were provided as supplementary information. However, based on your feedback, we recognize that this crucial piece of evidence may not have been sufficiently highlighted or elaborated upon in our manuscript. To address this, we have reanalyzed this piece of data and revised the manuscript to bring more visibility and emphasis to this data. In addition, we revised our discussion to remove the final statement indicating that TLNRD1 works independently from Rho/Rho kinase.

May 7, 2024

RE: JCB Manuscript #202310030R-A

Dr. Guillaume Jacquemet
Åbo Akademi University
Tykistökatu 6, 20520 Turku
Turku 20520
Finland

Dear Dr. Jacquemet:

Thank you for submitting your revised manuscript entitled "TLNRD1 is a CCM complex component and regulates endothelial barrier integrity". We would be happy to publish your paper in JCB pending final revisions necessary to meet our formatting guidelines (see details below), and addressing the remaining point raised by Reviewer 1.

A. MANUSCRIPT ORGANIZATION AND FORMATTING:

Full guidelines are available on our Instructions for Authors page, <http://jcb.rupress.org/submission-guidelines#revised>. Submission of a paper that does not conform to JCB guidelines will delay the acceptance of your manuscript.

1) Text limits: Character count for Articles is < 40,000, not including spaces. Count includes abstract, introduction, results, discussion, and acknowledgments. Count does not include title page, figure legends, materials and methods, references, tables, or supplemental legends.

2) Figures limits: Articles may have up to 10 main figures and 5 supplemental figures/tables.

** Please reduce supplemental figures. For instance Figure S7 may be included as a panel of Figure 6. In addition, one of the current supplemental figures may be made into a main figure.

3) Figure formatting: Scale bars must be present on all microscopy images, including inset magnifications. Molecular weight or nucleic acid size markers must be included on all gel electrophoresis. Please avoid pairing red and green for images and graphs to ensure legibility for color-blind readers. If red and green are paired for images, please ensure that the particular red and green hues used in micrographs are distinctive with any of the colorblind types. If not, please modify colors accordingly or provide separate images of the individual channels.

** Please include scale bars in Figure S5E.

4) Statistical analysis: Error bars on graphic representations of numerical data must be clearly described in the figure legend. The number of independent data points (n) represented in a graph must be indicated in the legend. Statistical methods should be explained in full in the materials and methods. For figures presenting pooled data the statistical measure should be defined in the figure legends. Please also be sure to indicate the statistical tests used in each of your experiments (either in the figure legend itself or in a separate methods section) as well as the parameters of the test (for example, if you ran a t-test, please indicate if it was one- or two-sided, etc.). Also, if you used parametric tests, please indicate if the data distribution was tested for normality (and if so, how). If not, you must state something to the effect that "Data distribution was assumed to be normal but this was not formally tested."

** Please specify the statistical test used in the legends for Figure 6B and C, and Figure S4D.

** Please indicate n in the legend for Figure S2 B.

5) Abstract and title: The abstract should be no longer than 160 words and should communicate the significance of the paper for a general audience. The title should be less than 100 characters including spaces. Make the title concise but accessible to a general readership.

6) Materials and methods: Should be comprehensive and not simply reference a previous publication for details on how an experiment was performed. Please provide full descriptions in the text for readers who may not have access to referenced manuscripts. We also provide a report from SciScore and an associate score, which we encourage you to use as a means of evaluating and improving the methods section.

7) Please be sure to provide the sequences for all of your primers/oligos and RNAi constructs in the materials and methods. You must also indicate in the methods the source, species, and catalog numbers (where appropriate) for all of your antibodies. Please also indicate the acquisition and quantification methods for immunoblotting/western blots.

8) Microscope image acquisition: The following information must be provided about the acquisition and processing of images:

- a. Make and model of microscope
- b. Type, magnification, and numerical aperture of the objective lenses
- c. Temperature
- d. Imaging medium
- e. Fluorochromes
- f. Camera make and model
- g. Acquisition software
- h. Any software used for image processing subsequent to data acquisition. Please include details and types of operations involved (e.g., type of deconvolution, 3D reconstitutions, surface or volume rendering, gamma adjustments, etc.).

10) Supplemental materials: There are strict limits on the allowable amount of supplemental data. Articles may have up to 5 supplemental figures. Please also note that tables, like figures, should be provided as individual, editable files. A summary of all supplemental material should appear at the end of the Materials and methods section.

13) ORCID IDs: ORCID IDs are unique identifiers allowing researchers to create a record of their various scholarly contributions in a single place. At resubmission of your final files, please provide an ORCID ID for all authors.

15) A data availability statement is required for all research article submissions. The statement should address all data underlying the research presented in the manuscript. Please visit the JCB instructions for authors for guidelines and examples of statements at (<https://rupress.org/jcb/pages/editorial-policies#data-availability-statement>).

Please note that JCB requires authors to submit Source Data used to generate figures containing gels and Western blots with all revised manuscripts. This Source Data consists of fully uncropped and unprocessed images for each gel/blot displayed in the main and supplemental figures. Since your paper includes cropped gel and/or blot images, please be sure to provide one Source Data file for each figure that contains gels and/or blots along with your revised manuscript files. File names for Source Data figures should be alphanumeric without any spaces or special characters (i.e., SourceDataF#, where F# refers to the associated main figure number or SourceDataFS# for those associated with Supplementary figures). The lanes of the gels/blots should be labeled as they are in the associated figure, the place where cropping was applied should be marked (with a box), and molecular weight/size standards should be labeled wherever possible. Source Data files will be directly linked to specific figures in the published article.

WHEN APPROPRIATE: The source code for all custom computational methods published in JCB must be made freely available as supplemental material hosted at www.jcb.org. Please contact the JCB Editorial Office to find out how to submit your custom macros, code for custom algorithms, etc. Generally, these are provided as raw code in a .txt file or as other file types in a .zip file. Please also include a one-sentence summary of each file in the Online Supplemental Material paragraph of your manuscript.

Journal of Cell Biology now requires a data availability statement for all research article submissions. These statements will be published in the article directly above the Acknowledgments. The statement should address all data underlying the research presented in the manuscript. Please visit the JCB instructions for authors for guidelines and examples of statements at (<https://rupress.org/jcb/pages/editorial-policies#data-availability-statement>).

B. FINAL FILES:

Thank you for your attention to these final processing requirements. Please revise and format the manuscript and upload materials within 7 days. If you need an extension for whatever reason, please let us know and we can work with you to determine a suitable revision period.

Thank you for this interesting contribution, we look forward to publishing your paper in Journal of Cell Biology.

Sincerely,

Tatiana Petrova
Monitoring Editor
Journal of Cell Biology

Tim Fessenden
Scientific Editor
Journal of Cell Biology

Reviewer #1 (Comments to the Authors (Required)):

The authors have adequately responded to the reviewer's comments, which significantly strengthened the manuscript. The work positions TLNRD1 as important new player in the CCM complex and endothelial barrier function.

There is one question that I would ask the authors to discuss: How come that a knockdown of TLNRD1 reduces endothelial barrier function, whereas at the same time the knockdown improves the recovery after thrombin-induced barrier loss? The finding is interesting, but it is also confusing and may well be explained by the notion that cells in the control condition somehow display a decline in barrier function long after the thrombin recovery.

Reviewer #2 (Comments to the Authors (Required)):

The authors found that the recently identified protein TLNRD1 is primarily expressed in the vasculature in vivo. Its depletion leads to vascular abnormalities in vivo and loss of barrier integrity in cultured endothelial cells. Furthermore, The authors show that TLNRD1 is part of the CMM complex and that biochemical precipitation assays identify CMM2 as a binding partner and link

this to vascular dementia.

Overall, this is a classic and thorough cell biology study. It shows how the protein interacts with a protein complex, and detects the localization in animal and cell models, using endogenous stainings and overexpression.

The authors addressed all comments I had on the Ms. It is solid research work.

Reviewer #1 (Comments to the Authors (Required)):

The authors have adequately responded to the reviewer's comments, which significantly strengthened the manuscript. The work positions TLNRD1 as an important new player in the CCM complex and endothelial barrier function.

We thank the reviewer for supporting the publication of our manuscript.

There is one question that I would ask the authors to discuss: How come that a knockdown of TLNRD1 reduces endothelial barrier function, whereas at the same time the knockdown improves the recovery after thrombin-induced barrier loss? The finding is interesting, but it is also confusing and may well be explained by the notion that cells in the control condition somehow display a decline in barrier function long after the thrombin recovery.

We thank the reviewer for raising this interesting point. The dual consequence of TLNRD1 silencing—impairing initial barrier formation yet enhancing recovery post-thrombin challenge—remains enigmatic. The reviewer's suggestion that control cells might exhibit a prolonged decline in barrier function is indeed a valid hypothesis. We propose that this paradox may stem from the intricate interplay between the TLNRD1-CCM2 interaction and TLNRD1's influence on the actin cytoskeleton. Given the dynamic nature of endothelial junction remodeling during both monolayer formation and recovery phases, further investigations are necessary to dissect these complex mechanisms and clarify TLNRD1's multifaceted role in endothelial barrier function.

In the manuscript, we write :

"Interestingly, our results show that TLNRD1 silencing initially delays the formation of an impermeable monolayer, yet it subsequently enhances recovery of barrier function following thrombin-induced damage. At this point, the dual roles of TLNRD1 remain elusive; however, we speculate that its complex functions may derive from its dynamic involvement in modulating the actin cytoskeleton and its contributions to the CCM complex. Future studies will focus on dissecting TLNRD1's roles during various endothelial barrier formation and function stages."

Reviewer #2 (Comments to the Authors (Required)):

The authors found that the recently identified protein TLNRD1 is primarily expressed in the vasculature in vivo. Its depletion leads to vascular abnormalities in vivo and loss of barrier integrity in cultured endothelial cells. Furthermore, The authors show that TLNRD1 is part of the CMM complex and that biochemical precipitation assays identify CMM2 as a binding partner and link this to vascular dementia.

Overall, this is a classic and thorough cell biology study. It shows how the protein interacts with a protein complex, and detects the localization in animal and cell models, using endogenous stainings and overexpression.

The authors addressed all comments I had on the Ms. It is solid research work.

We thank the reviewer for the kind word and supporting the publication of our manuscript.